# OSNIP: Balancing the Privacy-Utility-Efficiency Trilemma in LLM Inference via Obfuscated Semantic Null Space

Zhiyuan Cao [1 2]  Zeyu Ma [1 2 3]  Chenhao Yang [1 2 4]  Han Zheng [5]  Mingang Chen [1 2]

## Abstract

We propose Obfuscated Semantic Null Space Injection for Privacy (OSNIP), a lightweight client-side encryption framework for privacy-preserving LLM inference. Generalizing the geometric intuition of linear kernels to the high-dimensional latent space of LLMs, we formally define the "Obfuscated Semantic Null Space", a high-dimensional regime that preserves semantic fidelity while enforcing near-orthogonality to the original embedding. By injecting perturbations that project the original embedding into this space, OSNIP ensures privacy without any post-processing. Furthermore, OSNIP employs a key-dependent stochastic mapping that generates distinct perturbations under fresh keys. Evaluations on generative and classification benchmarks show that OSNIP achieves state-of-the-art performance, sharply reducing attack success rates while maintaining strong model utility under strict security constraints.

## 1. Introduction

The widespread adoption of Model-as-a-Service (MaaS) has democratized access to Large Language Models (LLMs) (Vaswani et al., 2017), allowing users to leverage powerful inference capabilities (M. Bran et al., 2024; Wu et al., 2023; Singhal et al., 2023) via cloud APIs. However, the centralized design of high-performance closed-source models requires sending user data to external model service providers, creating major privacy risks (Pan et al., 2020).

To mitigate these concerns, recent research[1] has explored various privacy-preserving inference strategies, primarily utilizing Homomorphic Encryption (HE) (de Castro et al., 2025; Hao et al., 2022), Differential Privacy (DP) (Wu et al., 2025; Chen et al., 2023; Yue et al., 2021; Zhang et al., 2025a), and Multi-Party Computation (MPC) (Li et al., 2024; Hou et al., 2023). These methods aim to protect user privacy while retaining model utility, following distinct paradigms: DP sanitizes inputs via perturbation, HE executes inference directly on encrypted ciphertexts, and MPC distributes computation shares across non-colluding parties.

Existing methods inherently view the high-dimensionality of LLM embeddings as a "curse", representing a landscape where computational overhead scales exponentially and standard cryptographic primitives become inapplicable. Consequently, these methods attempt to fight against dimensionality by imposing extrinsic constraints or altering the inference pipeline. Specifically, MPC requires architectural partitioning, HE requires invasive structural modifications, and text-substitution methods (like DP-based rewriting) often sacrifice semantic precision. These inevitably lead to a rigid trilemma: *a trade-off between inference efficiency, model utility, and privacy guarantees* (Zhang et al., 2025b).

However, we argue that the high dimensionality of LLMs is not a "curse", but a "blessing" for privacy. Motivated by recent works leveraging the null space for knowledge editing (Fang et al., 2025) and safety enhancement (Sheng et al., 2025), we demonstrate that the over-parameterization of LLMs induces an "**Obfuscated Semantic Null Space**", a high-dimensional region that preserves the output distribution while enforcing geometric decorrelation from the original embedding. Elements in this space are nearly orthogonal to the original embeddings while preserving semantic information. This ensures privacy and prevents inference interference without requiring denoising or reconstruction. By injecting perturbations that project the original embeddings into their obfuscated semantic null space, we effectively alleviate the aforementioned trade-offs to optimize efficiency, utility, and privacy in a single framework.

In this paper, we introduce Obfuscated Semantic Null Space Injection for Privacy (**OSNIP**), a lightweight client-side encryption framework. OSNIP uses geometric orthogonality as a primary privacy mechanism, employing a hinge-based

---

[1]Shanghai Key Laboratory of Computer Software Testing and Evaluating [2]Shanghai Development Center of Computer Software Technology [3]Shanghai Normal University [4]Shanghai Polytechnic University [5]TrustAI Pte. Ltd.. Correspondence to: Mingang Chen <cmg@sscenter.sh.cn>.

*Proceedings of the 43rd International Conference on Machine Learning*, Seoul, South Korea. PMLR 306, 2026. Copyright 2026 by the author(s).

[1]See Appendix A for additional related work.

constraint to minimize directional correlations. This procedure projects perturbations by constraining them to remain approximately orthogonal to the underlying semantic invariant region. Benefiting from the high-dimensional capacity of the obfuscated semantic null space, OSNIP further incorporates a key-dependent stochastic mapping that supports fresh-key randomized encryption. This mechanism ensures that the generated perturbation is uniquely conditioned on a secret key, enabling multiple distinct encryptions of the same input.

Our main contributions are as follows:

- **Obfuscated Semantic Null Space:** We formally define an "obfuscated semantic null space", representing a high-dimensional region that preserves the model's semantic distribution while remaining nearly orthogonal to the original embedding. This enables a novel denoising-free privacy protection paradigm, where perturbations can be injected without necessitating any post-processing.

- **OSNIP Framework:** We propose OSNIP, a client-side encryption framework that projects original embeddings into their obfuscated semantic null space via perturbation injection with negligible computational overhead. Specifically, using key-dependent stochastic mapping, OSNIP can generate distinct utility-preserving perturbations under different keys.

- **Superior Utility-Privacy Trade-off:** We conduct comprehensive evaluations across both generative and discriminative tasks on standard benchmarks, demonstrating that OSNIP maintains exceptional model utility even under stringent security constraints, effectively suppressing the Attack Success Rate (ASR) while preserving the original model's functional integrity. OSNIP establishes state-of-the-art (SOTA) performance in most evaluated scenarios.

**Conflict of Interest Disclosure.** The authors declare no financial conflicts of interest related to this work.

## 2. Problem Setup & Theoretical Analysis

### 2.1. Problem Setup

**System Setting.** We consider a Model-as-a-Service (MaaS) scenario involving three parties: a client $\mathcal{C}$, a cloud server $\mathcal{S}$ and a trusted third party $\mathcal{T}$. The client $\mathcal{C}$ holds a private prompt $x$ and secret key $k$, and seeks inference from an LLM hosted by $\mathcal{S}$ without revealing $x$ in plaintext. The server $\mathcal{S}$ hosts a pre-trained LLM $\mathcal{M}_\theta$ parameterized by $\theta$ and provides inference APIs. The trusted third party $\mathcal{T}$ retrieves the gradient of the LLM from server $\mathcal{S}$ to build an encryptor, subsequently deploying it to client $\mathcal{C}$.

Let $\mathcal{M}_\theta$ with a vocabulary $\mathcal{V}$ and embedding dimension $d$. Let $\mathcal{X}$, $\mathcal{Z} = \mathbb{R}^d$ and $\mathcal{Y} = \Delta^{|\mathcal{V}|}$ denote the input space, embedding space and output space, respectively. We decompose $\mathcal{M}_\theta$ into an embedding function $g : \mathcal{X} \to \mathcal{Z}$ and a downstream predictor $f_\theta : \mathcal{Z} \to \mathcal{Y}$. For an input $x$, its embedding representation is given by $\mathbf{h} = g(x)$.

Our goal is to design a perturbation mechanism $\mathcal{R}$ such that the perturbed embedding $\mathbf{z} = \mathcal{R}(h, k)$ maintains high utility for $\mathcal{M}_\theta$ while statistically hiding $x$ from $\mathcal{S}$.

**Adversary Setting.** We assume that the server $\mathcal{S}$ is a semi-honest adversary. The adversary strictly follows the inference protocol but logs all received embeddings to attempt to reconstruct the user's private input.

The server $\mathcal{S}$ possesses white-box access to the model parameters $\theta$ (including the vocabulary projection matrix) and is assumed to possess unbounded computational resources. Equipped with these capabilities, $\mathcal{S}$ can orchestrate geometry-dependent attacks, specifically K-Nearest Neighbors (Song & Raghunathan, 2020) (KNN) retrieval and vocabulary-matching attack (Thomas et al., 2025) via hidden states. However, $\mathcal{S}$ does not have access to the client's secret key $k$ and cannot tamper with the communication channel (assuming secure transmission).

### 2.2. Theoretical Analysis

In the context of textual privacy preservation for LLMs, we first introduce the definition of $d_\mathcal{X}$-privacy, following the framework established by Chatzikokolakis et al. (Chatzikokolakis et al., 2013).

Let $d_\mathcal{X}$ be the distance metric of $\mathcal{X}$. Let $\mathcal{P}(\mathcal{Y})$ denote the set of probability measures over the output space $\mathcal{Y}$. We formally define $d_\mathcal{X}$-privacy as follows:

**Definition 2.1** ($d_\mathcal{X}$-Privacy). A mechanism $K : \mathcal{X} \to \mathcal{P}(\mathcal{Y})$ satisfies $d_\mathcal{X}$-privacy iff for all $x, x' \in \mathcal{X}$:

$$d_\mathcal{P}(K(x), K(x')) \le d_\mathcal{X}(x, x'), \tag{1}$$

where $d_\mathcal{P}(\cdot, \cdot)$ is the multiplicative distance between two distributions.

Definition 2.1 extends privacy guarantees to the continuous geometry of LLM embeddings. To identify a regime where we can maximize input distortion (privacy) while minimizing output divergence (utility), consider the trivial case where the mechanism K is a linear transformation. In this setting, the condition is perfectly satisfied when the perturbation is chosen from the null directions that are also orthogonal to the input, i.e. $\ker(K) \cap x^\perp$, so the output distance $d_\mathcal{P}$ stays zero while the perturbed input becomes geometrically decorrelated from $x$.

Departing from idealized linear systems, LLMs implement highly non-linear mappings, so a global null space is no

longer available in closed form. Instead, we appeal to the manifold hypothesis (Bengio et al., 2013): embeddings of natural language inputs concentrate near a low-dimensional semantic manifold. Around a reference embedding $\mathbf{h} \in \mathcal{M}$, the predictor $f_\theta$ typically exhibits many near-invariant directions, yielding a local neighborhood in which the predictive distribution changes only marginally. Crucially, such near-invariance is input-conditioned—it depends on the local geometry around $\mathbf{h}$ rather than defining a global subspace. We formalize this space as the following semantic null space.

**Definition 2.2** (Semantic Null Space). Given an LLM predictor f, a reference embedding $\mathbf{h}$ and tolerance $\delta > 0$, the *semantic null space* is defined as

$$\mathcal{S}_\delta^{\mathrm{f}}(\mathbf{h}) \; = \; \Big\{ \mathbf{z} \in \mathbb{R}^d \;\big|\; d_\mathcal{P}\big(f_\theta(\mathbf{h}), f_\theta(\mathbf{z})\big) \le \delta \Big\}. \quad (2)$$

Elements of $\mathcal{S}_\delta^{\mathrm{f}}(\mathbf{h})$ induce predictive distributions that are $\delta$-close to that of $\mathbf{h}$ with respect to the distance measure $d_\mathcal{P}$ for an LLM predictor $f_\theta$.

While $\mathcal{S}_\delta^{\mathrm{f}}(\mathbf{h})$ characterizes the constraints required for utility, it does not inherently guarantee privacy. To defend against geometry-dependent adversaries, the perturbed embedding must be geometrically distant from the original input. Complementing the semantic constraint, we define a *geometric obfuscation region* that enforces low directional similarity to the reference embedding, thereby suppressing cosine/KNN-style matching signals exploited by geometry-based inversion attacks.

**Definition 2.3** (Geometric Obfuscation Region). For a reference embedding $\mathbf{h}$ and an orthogonality margin $\epsilon \ge 0$, the *geometric obfuscation region* $\mathcal{O}_\epsilon(\mathbf{h}) \subset \mathbb{R}^d$ is defined as:

$$\mathcal{O}_\epsilon(\mathbf{h}) = \Big\{ \mathbf{z} \in \mathcal{Z} \,\big|\, |\cos(\mathbf{h}, \mathbf{z})| \le \epsilon \Big\}. \quad (3)$$

Geometrically, this set constitutes a hyperspherical band (see Figure 1) centered at the equator relative to the pole $\mathbf{h}$. Vectors within this locus are statistically orthogonal to the input, rendering distance-based inversion attacks ineffective.

Recall our initial motivation from the linear case: we seek a subspace where perturbed inputs are significant (high $d_\mathcal{X}$) yet the output remains invariant (zero or low $d_\mathcal{P}$). Clearly, provided that the intersection between the safety region $S$ and the objective set $O$ is non-empty, every element $\delta \in S \cap O$ serves as a qualified perturbed embedding. Specifically, vectors within this intersection possess a dual nature: they are geometrically orthogonal to the input yet semantically equivalent to the LLM. Crucially, this implies that the injected noise *does not need to be removed*; it is inherently "ignored" by the LLM's semantic projection. We define this intersection as the Obfuscated Semantic Null Space.

**Definition 2.4** (Obfuscated Semantic Null Space). Given an LLM predictor $f(\cdot)$, a reference embedding $\mathbf{h}$, a utility

tolerance $\delta$, and an orthogonality margin $\epsilon$, the obfuscated semantic null space $\mathcal{N}_{\delta,\epsilon}(\mathbf{h})$ is defined as the intersection of the semantic invariance region and the $\epsilon$-orthogonal region:

$$\mathcal{N}_{\delta,\epsilon}^{\mathrm{f}}(\mathbf{h}) := \mathcal{S}_\delta^{\mathrm{f}}(\mathbf{h}) \cap \mathcal{O}_\epsilon(\mathbf{h}). \quad (4)$$

For notational simplicity, we denote it by $\mathcal{N}_{\delta,\epsilon}(\mathbf{h})$.

As long as $\mathcal{N} \ne \varnothing$, it allows for the injection of perturbations that are functionally invariant to the LLM's predictive logic, thereby achieving privacy preservation without the need for subsequent reconstruction.

**Theoretical Results.** To avoid ambiguities of unbounded Lebesgue volume, we measure volumes on the sphere of radius $r := \|\mathbf{h}\|_2$:

$$\mathbb{S}_r^{d-1} := \{\mathbf{z} \in \mathbb{R}^d : \|\mathbf{z}\|_2 = r\}. \quad (5)$$

Let $\mu_{d,r}$ denote the uniform probability measure on $\mathbb{S}_r^{d-1}$.

We define the set of *semantic-preserving directions* as

$$\Omega_\delta(\mathbf{h}) := \Big\{ \mathbf{u} \in \mathbb{S}_1^{d-1} \;\big|\; d_\mathcal{P}\big(f_\theta(\mathbf{h}), f_\theta(r\mathbf{u})\big) \le \delta \Big\}. \quad (6)$$

Accordingly, we define the *semantic coverage rate*

$$\alpha_\delta(\mathbf{h}) := \sigma_{d-1}\big(\Omega_\delta(\mathbf{h})\big) \in [0, 1]. \quad (7)$$

Intuitively, $\alpha_\delta(\mathbf{h})$ measures the fraction of iso-norm directions that preserve the predictive distribution within tolerance $\delta$.

We next formalize the geometric obfuscation constraint on the same compact geometry. Let $\widehat{\mathbf{h}} := \mathbf{h}/\|\mathbf{h}\|_2$ be the normalized reference vector. We consider the restriction of the $\epsilon$-orthogonality constraint to the unit sphere as

$$\mathcal{B}_\epsilon(\mathbf{h}) := \Big\{ \mathbf{u} \in \mathbb{S}_1^{d-1} \;\big|\; |\langle \mathbf{u}, \widehat{\mathbf{h}} \rangle| \le \epsilon \Big\}. \quad (8)$$

Consequently, the *directional obfuscated semantic null space* is defined as the intersection of the semantic-preserving directions and the spherical $\epsilon$-orthogonal band:

$$\begin{aligned} \mathcal{N}_{\delta,\epsilon}^{\mathrm{dir}}(\mathbf{h}) &:= \Omega_\delta(\mathbf{h}) \cap \mathcal{B}_\epsilon(\mathbf{h}), \\ \mathcal{N}_{\delta,\epsilon}(\mathbf{h}) &= \{r\mathbf{u} : \mathbf{u} \in \mathcal{N}_{\delta,\epsilon}^{\mathrm{dir}}(\mathbf{h})\}. \end{aligned} \quad (9)$$

**Theorem 2.5** (Existence of Semantic Null Space). *Suppose the semantic coverage rate satisfies the condition:*

$$\alpha_\delta(\mathbf{h}) > 2\exp\Big(-\frac{(d-2)\epsilon^2}{2}\Big). \quad (10)$$

*Then $\mathcal{N}_{\delta,\epsilon}^{\mathrm{dir}}(\mathbf{h}) \ne \varnothing$, hence $\mathcal{N}_{\delta,\epsilon}(\mathbf{h}) \ne \varnothing$. Moreover,*

$$\sigma\big(\mathcal{N}_{\delta,\epsilon}^{\mathrm{dir}}(\mathbf{h})\big) \ge \alpha_\delta(\mathbf{h}) - 2\exp\Big(-\frac{(d-2)\epsilon^2}{2}\Big) > 0. \quad (11)$$

Theorem 2.5 formalizes our core geometric claim: high dimensionality makes semantic null space ubiquitous. Specifically, the complement of the $\epsilon$-orthogonal band occupies only $2\exp\left(-\frac{(d-2)\epsilon^2}{2}\right)$ spherical mass, which vanishes exponentially fast as $d$ grows. Therefore, as long as the semantic-preserving directions have non-negligible coverage $\alpha_\delta(\mathbf{h})$, their intersection with the orthogonal band is guaranteed to be non-empty.

**Corollary 2.6** (Asymptotic Semantic Dominance)**.** *For any* $\delta > 0$ *and* $\epsilon \in (0, 1)$*, we have*

$$0 \;\leq\; \alpha_\delta(\mathbf{h}) - \sigma_{d-1}\big(\mathcal{N}_{\delta,\epsilon}^{\mathrm{dir}}(\mathbf{h})\big) \;\leq\; 2\exp\Big(-\frac{(d-2)\epsilon^2}{2}\Big). \tag{12}$$

*In particular, the gap vanishes exponentially fast in d, i.e.,* $\alpha_\delta(\mathbf{h}) - \sigma_{d-1}\big(\mathcal{N}_{\delta,\epsilon}^{\mathrm{dir}}(\mathbf{h})\big) \to 0$ *as* $d \to \infty$*.*

Corollary 2.6 shows that, as $d$ grows, the orthogonality constraint becomes asymptotically non-binding: the semantic null space occupies nearly the same spherical mass as the semantic-preserving set, with a gap upper bounded by $2\exp\left(-\frac{(d-2)\epsilon^2}{2}\right)$.

# 3. Obfuscated Semantic Null Space Injection for Privacy (OSNIP)

## 3.1. Overview

Section 2.2 establishes that, under mild conditions, the semantic null space $\mathcal{N}_{\delta,\epsilon}(\mathbf{h})$ is non-empty and typically of non-negligible measure in high dimensions. We now move from theory to practice.

**OSNIP** (Obfuscated Semantic Null Space Injection for Privacy) is a lightweight client-side encryption framework that transforms a clean embedding $\mathbf{h}$ into an *encrypted embedding* $\mathbf{z}$, such that (i) the cloud-hosted LLM exhibits nearly identical predictive behavior, and (ii) the resulting representation is geometrically uninformative for inversion attacks that rely on embedding proximity. Crucially, OSNIP enforces invariance *at the source*: the perturbation is injected along directions that the model is intrinsically insensitive to, eliminating the need for any denoising, reconstruction, or server-side modification.

Figure 1 illustrates the end-to-end workflow. First, $\mathcal{T}$ trains an encryption network using gradients from the server-side LLM and deploys it to the client $\mathcal{C}$. At inference time, the client $\mathcal{C}$ computes the clean embedding $\mathbf{h}$ and feeds $(\mathbf{h}, k)$ into the encryption network, which outputs an encrypted embedding $\mathbf{z} = \mathcal{R}(\mathbf{h}, k)$. The client then sends $\mathbf{z}$ to the cloud server, which performs standard inference without any architectural partitioning or cryptographic protocol.

A key feature of OSNIP is secret-key-conditioned randomization. Conditioning on a secret key $k$ makes the mapping from $\mathbf{h}$ to $\mathbf{z}$ inherently non-deterministic: the same prompt can yield different encrypted embeddings under different fresh keys. This key conditioning makes the encrypted embeddings less linkable across clients and interactions, and limits an attacker's ability to train a single universal inverter from logged embeddings.

In the following subsections, we detail the OSNIP framework. Section 3.2 introduces a lightweight encryption network and its training objective, which injects perturbations to project the original embeddings into their obfuscated semantic null space. Section 3.3 then presents key-conditioned randomization, where an added regularizer promotes key-specific diversity. Finally, Section 3.4 outlines a dynamic training strategy for stabilizing this multi-objective optimization, ensuring that utility preservation, orthogonality, and key-conditioned diversity can be satisfied simultaneously in practice.

## 3.2. Encryption Network

Given a clean embedding $\mathbf{h} = \mathrm{g}(x)$, the encryptor produces an encrypted embedding

$$\mathbf{z} = \mathcal{R}_\phi(\mathbf{h}), \tag{13}$$

which is then sent to the cloud server for standard inference. The server-side model parameters $\theta$ remain unchanged throughout training and deployment.

**Design principle.** Our goal is to make $\mathbf{z}$ satisfy two requirements simultaneously: (i) utility preservation—the server-side predictor should produce nearly identical output distributions when fed with $\mathbf{h}$ or $\mathbf{z}$; (ii) geometric obfuscation—$\mathbf{z}$ should be directionally decorrelated from $\mathbf{h}$ to suppress cosine/KNN-style matching signals. Instead of explicitly solving for $\mathbf{z} \in \mathcal{N}_{\delta,\epsilon}(\mathbf{h})$, we enforce these two requirements via a simple multi-objective training loss.

**Utility loss.** We instantiate the distribution distance $d_{\mathcal{P}}(\cdot, \cdot)$ using the KL divergence between the predictive distributions. We define the utility loss as

$$\mathcal{L}_{\mathrm{util}} \;=\; D_{\mathrm{KL}}\big(\mathrm{f}_\theta(\mathbf{h}) \,\|\, \mathrm{f}_\theta(\mathbf{z})\big). \tag{14}$$

**Privacy loss.** To enforce geometric obfuscation, we penalize directional similarity between $\mathbf{h}$ and $\mathbf{z}$ using a hinge loss:

$$\mathcal{L}_{\mathrm{priv}} \;=\; \max\big(0, |\cos(\mathbf{h}, \mathbf{z})| - \epsilon\big), \tag{15}$$

where $\cos(\mathbf{h}, \mathbf{z}) = \frac{\langle \mathbf{h}, \mathbf{z} \rangle}{\|\mathbf{h}\|_2 \|\mathbf{z}\|_2}$ and $\epsilon \in [0, 1)$ controls the target orthogonality margin. This hinge form avoids over-penalizing already-obfuscated embeddings and stabilizes optimization.

**Overall training objective.** We train the encryptor parameters $\phi$ by minimizing a weighted sum of the two losses:

$$\min_\phi \; \mathbb{E}_{x \sim \mathcal{D}}\Big[\mathcal{L}_{\mathrm{util}} \;+\; \lambda \, \mathcal{L}_{\mathrm{priv}}\Big], \tag{16}$$

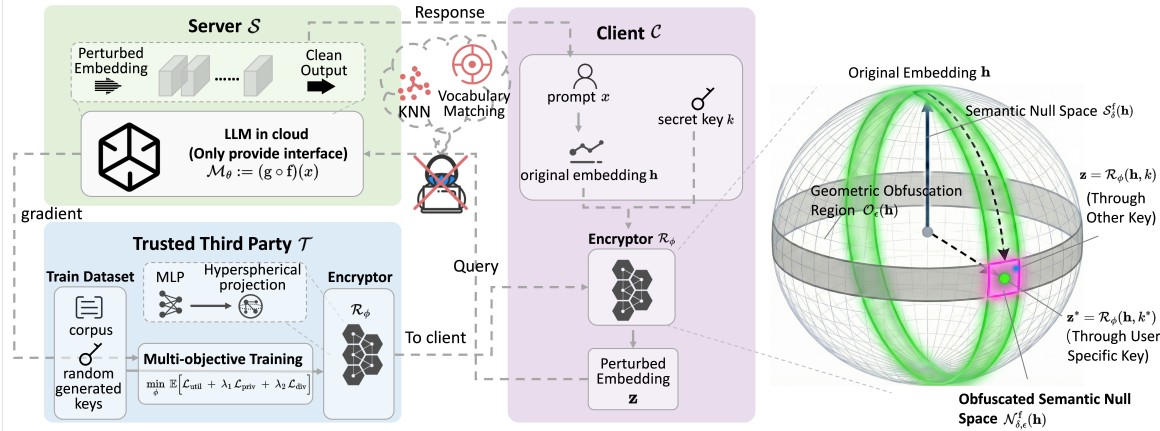

*Figure 1.* The OSNIP Architecture. Trusted Third Party trains an encryptor using corpora and randomized keys before deploying it client-side. Using their prompts and private keys, clients generate perturbed embeddings that resist privacy attacks by servers or interceptors, while still enabling standard server-side inference.

where $\lambda > 0$ balances utility preservation and geometric obfuscation. In practice, we freeze $\theta$ and backpropagate gradients through the server-side predictor $f_\theta$ to update $\phi$. This procedure steers the encryptor to output embeddings $\mathbf{z}$ that are simultaneously utility-preserving and directionally separated from $\mathbf{h}$, thereby injecting perturbations to project the original embeddings into their obfuscated semantic null space defined in Section 2.2.

**Training and deployment protocol.** A trusted third party $\mathcal{T}$ performs the above optimization using gradient access to the server-side LLM. After training, $\mathcal{T}$ deploys the encryptor $\mathcal{R}_\phi$ to the client $\mathcal{C}$. At inference time, the client computes $\mathbf{h} = g(x)$, applies $\mathcal{R}_\phi$ to obtain $\mathbf{z}$, and sends $\mathbf{z}$ to the server. The cloud server then runs the original inference pipeline on $\mathbf{z}$ without any architectural change, partitioning, or cryptographic protocol.

**Hyperspherical energy projection.** Standard perturbation can alter the magnitude of vectors, thereby disrupting the distribution of dot-products in subsequent Self-Attention layers. To preserve inference stability and adhere to the geometry of the pre-trained manifold, we enforce an Iso-Norm Constraint by projecting $\mathbf{z}$ back onto the hypersphere $\mathbb{S}_{\|h\|}$ of the original radius:

$$\tilde{\mathbf{z}} = \text{Project}(\mathbf{z}, \mathbf{h}) = (\mathbf{h} + \delta) \cdot \frac{\|\mathbf{h}\|_2}{\|\mathbf{h} + \delta\|_2} \quad (17)$$

This operation ensures that the obfuscation is strictly directional, decoupling the semantic magnitude from the geometric orientation, which aligns with the characteristics of our adversary setting modeled in Section 2.1.

### 3.3. Key-Conditioned Randomization

Section 3.2 introduces a base encryptor that maps a clean embedding $\mathbf{h}$ to an encrypted embedding $\mathbf{z}$ while preserving

utility and enforcing geometric obfuscation. We now extend OSNIP with key-conditioned randomization, enabling distinct (yet utility-preserving) encryptions for the same input under different fresh keys. This randomization reduces cross-interactions linkability in embedding logs and limits an attacker's ability to train a single universal inverter that generalizes across different keys. **Key-dependent encryptor.** We augment the encryptor (Equation (13)) with a secret key $k$:

$$\mathbf{z} = \mathcal{R}_\phi(\mathbf{h}, k). \quad (18)$$

Intuitively, conditioning on $k$ allows OSNIP to generate multiple valid encrypted embeddings for the same $\mathbf{h}$, which reduces cross-interactions linkability and mitigates attacks.

**Key diversity regularization.** A practical failure mode is that the network may ignore $k$ and collapse to a key-agnostic mapping. To prevent this, we explicitly encourage *geometric separation* between encrypted outputs produced under different keys. For the same $\mathbf{h}$, we sample two keys $k_1 \neq k_2$ and enforce a margin in Euclidean distance via a hinge loss:

$$\mathcal{L}_{\text{div}} = \max\left(0, \, \delta - \left\|\mathcal{R}_\phi(\mathbf{h}, k_1) - \mathcal{R}_\phi(\mathbf{h}, k_2)\right\|_2\right), \quad (19)$$

where $\delta > 0$ is a separation margin. Unlike unbounded maximization, the hinge form stops penalizing once the distance exceeds $\delta$, which stabilizes training and avoids over-dispersing the encrypted embeddings.

**Overall objective with keys.** With key conditioning, the training objective becomes

$$\min_{\phi} \, \mathbb{E}_{x \sim \mathcal{D}, \, k_1 \neq k_2}\left[\mathcal{L}_{\text{util}} + \lambda_1 \mathcal{L}_{\text{priv}} + \lambda_2 \mathcal{L}_{\text{div}}\right], \quad (20)$$

where $\lambda_1 > 0$ controls geometric obfuscation and $\lambda_2 > 0$ controls the strength of key-conditioned diversity. Sec-

tion 3.4 details a dynamic optimization strategy for balancing these objectives in practice.

### 3.4. Optimization Strategy: Utility-Gated Curriculum

Training OSNIP involves a multi-objective optimization that simultaneously enforces (i) semantic consistency, (ii) geometric obfuscation, and (iii) key-conditioned diversity. Directly optimizing the objectives can cause semantic rigidity: overly strong geometric constraints may push the encrypted representation out of the semantic null space before the encryptor learns stable, utility-preserving directions.

To mitigate this, we introduce a utility-gated curriculum learning strategy. Instead of a static schedule, the constraint weights $\lambda(t)$ are dynamically modulated by two factors: a time-based warmup and a performance-based safety gate.

$$\lambda(t, \mathcal{L}_{\text{util}}) = \lambda_{\text{base}} \cdot \underbrace{w_{\text{time}}(t)}_{\text{Warmup}} \cdot \underbrace{w_{\text{safe}}(\mathcal{L}_{\text{util}})}_{\text{Safety Gate}} \quad (21)$$

**Time-based Warmup** ($w_{\text{time}}$)**:** During the initial phase, we linearly increase the penalty from 0 to 1 during the initial phase (e.g., first 1k steps).

**Utility-Gated Safety Mechanism** ($w_{\text{safe}}$)**:** Warmup alone is insufficient when the privacy gradients occasionally dominate and destabilize utility. We therefore introduce a closed-loop gate based on the instantaneous utility loss. Given a predefined safe interval $[\tau_{\text{low}}, \tau_{\text{high}}]$, the gate smoothly down-weights constraints when utility degrades:

$$w_{\text{safe}}(\ell) \ = \ \text{clip}\left(\frac{\tau_{\text{high}} - \ell}{\tau_{\text{high}} - \tau_{\text{low}}}, 0, 1\right). \quad (22)$$

When $\mathcal{L}_{\text{util}}$ rises above the safe range, the gate temporarily relaxes the privacy constraints, allowing the optimization to re-establish semantic consistency before re-applying stronger obfuscation and diversity pressure. Empirically, this utility-gated curriculum makes the three objectives compatible and yields stable convergence without sacrificing the denoising-free inference property of OSNIP. Appendix D.6 reports the default hyperparameters and scale-dependent selection rules.

## 4. Experiments

### 4.1. Experimental Setup

**Models.** We evaluate OSNIP on four open-source LLMs: Llama-3.2-1B, Llama-3.2-3B-Instruct (Grattafiori et al., 2024), Qwen3-14B and Qwen3-32B (Yang et al., 2025).

**Baselines.** We compare against Cape (Wu et al., 2025), DYNTEXT (Zhang et al., 2025a), and InferDPT (Tong et al., 2025), using official/default configurations. For each baseline, we adopt the privacy budget settings reported to yield

the best utility–privacy trade-off in their respective papers or official implementations.

**Training Data.** We train the encryptor on a 65k-sample mixture: 80% general instructions (Alpaca (Taori et al., 2023)) and 20% science/reasoning (ARC-Easy/Challenge (Clark et al., 2018) and SciQ (Welbl et al., 2017)).

**Benchmarks.** We evaluate both closed- and open-ended tasks. *Closed-ended:* MMLU (Hendrycks et al., 2021), ARC-Easy (Clark et al., 2018), HellaSwag (Zellers et al., 2019), PIQA (Bisk et al., 2020), MNLI (Williams et al., 2018), SST2 (Socher et al., 2013), ANLI (R1/R2/R3) (Nie et al., 2020), and WiC (Pilehvar & Camacho-Collados, 2019). *Open-ended:* (i) text completion on WikiText-2 (Merity et al., 2017) (PPL with a sliding window), and (ii) summarization on 1,000 CNN/DailyMail (Hermann et al., 2015) test samples (ROUGE-L (Lin, 2004) and BERTScore (Zhang et al., 2020)).*High-precision:* we additionally evaluate coding and math utility on HumanEval (Chen et al., 2021) using Pass@1 and GSM8k (Cobbe et al., 2021) using accuracy. These supplementary tests are reported for Qwen3-14B and Qwen3-32B.

**Privacy Metrics.** We simulate a semi-honest server and report ASR under two embedding-based attacks. *KNN attack*: the attacker ranks all vocabulary tokens by Euclidean distance to the perturbed token embedding and checks whether the ground-truth token appears in the Top-$k$ list (we report $k \in \{1, 5, 10\}$). *Vocabulary-matching attack*: the attacker reconstructs the sequence autoregressively by selecting, at each step, the vocabulary token whose hidden state is closest (in $L_1$ distance) to the perturbed representation, and feeds the output back for the next step.

### 4.2. Main Results

**Superior utility preservation across scales.** As shown in Table 1, OSNIP achieves near-lossless utility retention on standard benchmarks. This advantage extends to generative tasks: Table 2 reveals that baselines suffer from an "Explosion", while OSNIP maintains a stable, low-perplexity profile consistent with the original model. Crucially, for the summarization task shown in Table 3, OSNIP achieves near-lossless semantic preservation. Unlike InferDPT, which depends on reconstruction to recover utility, OSNIP preserves this high fidelity in a fully end-to-end manner.

**The "dimensionality dividend".** Regarding privacy, OSNIP achieves near "zero-leakage" privacy, while baselines cannot reduce ASR below 0.3 without severely harming utility. We observe a distinct positive scalability for OSNIP: as model size grows, its privacy-utility trade-off improves. We attribute this to high-dimensional representations: as model dimension grows, the Obfuscated Semantic Null Space expands, providing exponentially more freedom to find an

*Table 1.* Main Results on Privacy-Utility Trade-off. Privacy is measured by KNN-Top10 Attack Success Rate (ASR ↓) in Section 4.4. Utility is measured by accuracy on 10 benchmarks (↑). Avg denotes the average accuracy across all 10 tasks. RP (Retained Performance) indicates the percentage of utility preserved relative to the Non-private baseline. For baselines, we adopt the hyperparameter settings that yield their best performance for comparison.

| MODEL | METHOD | CONF. | PRIV. ASR ↓ | UTILITY METRICS (ACCURACY ↑) | | | | | | | | | | SUMMARY | |
|---|---|---|---|---|---|---|---|---|---|---|---|---|---|---|---|
| | | | | ARC-E | PIQA | MNLI | SST2 | ANLI1 | ANLI2 | ANLI3 | WiC | HELLAS[†] | MMLU[†] | AVG | RP |
| QWEN3-32B | NON-PRIVATE | / | 1.000 | 0.830 | 0.819 | 0.699 | 0.929 | 0.691 | 0.616 | 0.589 | 0.657 | 0.826 | 0.818 | 0.747 | 100% |
| | CAPE | $\epsilon=14$ | 0.578 | 0.593 | 0.693 | 0.531 | 0.747 | 0.511 | 0.459 | 0.468 | 0.539 | 0.340 | 0.339 | 0.522 | 69.84% |
| | DYNTEXT | $\epsilon=2.5$ | 0.761 | 0.480 | 0.552 | 0.394 | 0.530 | 0.396 | 0.378 | 0.398 | 0.520 | 0.394 | 0.238 | 0.428 | 57.27% |
| | INFERDPT | $\epsilon=6$ | 0.622 | 0.265 | 0.496 | 0.338 | 0.493 | 0.333 | 0.328 | 0.336 | 0.506 | 0.279 | 0.249 | 0.362 | 48.44% |
| | **OSNIP (OURS)** | / | **0.000** | **0.830** | **0.814** | **0.698** | **0.929** | **0.702** | **0.612** | **0.593** | **0.665** | **0.825** | **0.812** | **0.748** | **100.13%** |
| QWEN3-14B | NON-PRIVATE | / | 1.000 | 0.830 | 0.799 | 0.670 | 0.924 | 0.665 | 0.574 | 0.557 | 0.577 | 0.788 | 0.787 | 0.717 | 100% |
| | CAPE | $\epsilon=14$ | 0.334 | 0.395 | 0.563 | 0.374 | 0.567 | 0.332 | 0.345 | 0.339 | 0.531 | 0.317 | 0.253 | 0.402 | 56.07% |
| | DYNTEXT | $\epsilon=2.5$ | 0.745 | 0.456 | 0.527 | 0.355 | 0.523 | 0.366 | 0.362 | 0.398 | 0.527 | 0.375 | 0.240 | 0.413 | 57.58% |
| | INFERDPT | $\epsilon=6$ | 0.616 | 0.267 | 0.508 | 0.339 | 0.502 | 0.337 | 0.314 | 0.316 | 0.498 | 0.274 | 0.247 | 0.360 | 50.21% |
| | **OSNIP (OURS)** | / | **0.000** | **0.827** | **0.792** | **0.649** | **0.921** | **0.641** | **0.541** | **0.548** | **0.589** | **0.787** | **0.787** | **0.708** | **98.76%** |
| LLAMA-3.2-3B-INSTRUCT | NON-PRIVATE | / | 1.000 | 0.712 | 0.768 | 0.542 | 0.867 | 0.440 | 0.401 | 0.425 | 0.497 | 0.716 | 0.606 | 0.597 | 100% |
| | CAPE | $\epsilon=14$ | 0.404 | 0.490 | 0.613 | 0.461 | 0.642 | 0.366 | 0.345 | 0.365 | 0.486 | 0.324 | 0.245 | 0.434 | 72.70% |
| | DYNTEXT | $\epsilon=2.5$ | 0.807 | 0.266 | 0.517 | 0.321 | 0.511 | 0.349 | 0.339 | 0.348 | 0.480 | 0.275 | 0.243 | 0.365 | 61.14% |
| | INFERDPT | $\epsilon=6$ | 0.667 | 0.269 | 0.509 | 0.331 | 0.508 | 0.352 | 0.320 | 0.333 | 0.491 | 0.271 | 0.239 | 0.362 | 60.65% |
| | **OSNIP (OURS)** | / | **0.021** | **0.707** | **0.766** | **0.535** | **0.825** | **0.431** | **0.407** | **0.410** | **0.513** | **0.709** | **0.572** | **0.588** | **98.49%** |
| LLAMA-3.2-1B | NON-PRIVATE | / | 1.000 | 0.617 | 0.748 | 0.358 | 0.580 | 0.349 | 0.331 | 0.349 | 0.445 | 0.642 | 0.314 | 0.473 | 100% |
| | CAPE | $\epsilon=14$ | 0.410 | 0.449 | 0.615 | 0.352 | 0.587 | 0.339 | 0.327 | 0.324 | 0.461 | 0.310 | 0.230 | 0.399 | 84.36% |
| | DYNTEXT | $\epsilon=2.5$ | 0.802 | 0.265 | 0.522 | 0.327 | 0.490 | 0.325 | 0.320 | 0.333 | 0.487 | 0.266 | 0.264 | 0.360 | 76.11% |
| | INFERDPT | $\epsilon=6$ | 0.661 | 0.263 | 0.498 | 0.346 | 0.518 | **0.353** | 0.329 | 0.310 | **0.511** | 0.269 | 0.238 | 0.364 | 76.96% |
| | **OSNIP (OURS)** | / | **0.066** | **0.603** | **0.719** | **0.358** | **0.733** | 0.337 | **0.336** | **0.333** | 0.481 | **0.615** | **0.282** | **0.480** | **101.48%** |

optimal perturbation **z** that both strengthens privacy and preserves inference performance. In contrast, baselines exhibit inverse scalability, where the "safe" replacement region narrows in larger models, leading to the observed utility collapse on larger models.

**Mechanism Analysis: Continuous vs. Discrete.** The performance gap highlights the fragility of discrete token perturbation: even near-synonym substitutions can break the cues that support multi-step reasoning. OSNIP avoids this by operating in the continuous embedding space. By utilizing the obfuscated semantic null space, OSNIP protects privacy without compromising the model's intrinsic reasoning or generative capabilities. Additional evaluations on coding and math tasks in Section 4.3 show that OSNIP remains competitive on high-precision tasks while keeping KNN ASR low.

### 4.3. Math and Coding Benchmarks

To assess utility on high-precision tasks, we evaluate OSNIP on HumanEval (Chen et al., 2021) and GSM8k (Cobbe et al., 2021). These tasks are sensitive to small semantic shifts because code generation and mathematical reasoning require exact logical consistency. Table 4 shows that OSNIP preserves competitive performance on both tasks while keeping the average KNN ASR at 0.

*Table 2.* Perplexity (PPL) evaluation on WikiText-2. NON-P denotes the non-private baseline, and PRIV. refers to the private methods. $\Delta\%$ indicates the percentage increase in PPL.

| MODEL | METHOD | CONF. | NON-P | PRIV. (↓) | Δ (% ↓) |
|---|---|---|---|---|---|
| QWEN3-32B | CAPE | $\epsilon=14$ | 6.73 | 62.70 | +832% |
| | DYNTEXT | $\epsilon=2.5$ | 6.73 | 53.94 | +701% |
| | INFERDPT | $\epsilon=6$ | 6.73 | 1155.74 | +17073% |
| | **OSNIP** | / | 6.73 | **6.94** | **+3%** |
| QWEN3-14B | CAPE | $\epsilon=14$ | 7.59 | 67.55 | +790% |
| | DYNTEXT | $\epsilon=2.5$ | 7.59 | 58.48 | +670% |
| | INFERDPT | $\epsilon=6$ | 7.59 | 1184.29 | +15503% |
| | **OSNIP** | / | 7.59 | **7.69** | **+1%** |
| LLAMA-3.2-3B-INSTRUCT | CAPE | $\epsilon=14$ | 9.63 | 97.56 | +913% |
| | DYNTEXT | $\epsilon=2.5$ | 9.63 | 112.24 | +1066% |
| | INFERDPT | $\epsilon=6$ | 9.63 | 2869.66 | +29699% |
| | **OSNIP** | / | 9.63 | **12.58** | **+31%** |
| LLAMA-3.2-1B | CAPE | $\epsilon=14$ | 8.63 | 80.58 | +834% |
| | DYNTEXT | $\epsilon=2.5$ | 8.63 | 82.22 | +853% |
| | INFERDPT | $\epsilon=6$ | 8.63 | 2074.80 | +23942% |
| | **OSNIP** | / | 8.63 | **11.27** | **+31%** |

### 4.4. Defense Against Attacks

We evaluate the robustness of OSNIP against two distinct inversion attacks: the KNN attack and the vocabulary-matching attack. As shown in our evaluation, while baseline methods remain vulnerable to these inversion attacks, OSNIP achieves SOTA defense against the KNN attack and maintains a low ASR against the vocabulary-matching attack. (Detailed in Appendix C)

*Table 3.* Text summarization performance on CNN/DailyMail. Evaluation metrics include ROUGE-L and BERTScore. Non-P denotes the non-private baseline, and RP (Retained Performance) indicates the percentage of BERTScore preserved relative to the baseline.

| MODEL | METHOD | ROUGE-L (↑) | | BERTSCORE (↑) | | |
|---|---|---|---|---|---|---|
| | | NON-P | PRIV. | NON-P | PRIV. | RP |
| QWEN3-32B | CAPE | 0.183 | 0.150 | 0.866 | 0.855 | 98.7% |
| | DYNTEXT | 0.183 | 0.118 | 0.866 | 0.836 | 96.5% |
| | INFERDPT | 0.183 | **0.197** | 0.866 | **0.866** | **100.0%** |
| | **OSNIP** | 0.183 | 0.182 | 0.866 | 0.865 | **99.9%** |
| QWEN3-14B | CAPE | 0.163 | 0.132 | 0.852 | 0.840 | 98.6% |
| | DYNTEXT | 0.163 | 0.117 | 0.852 | 0.837 | 98.2% |
| | INFERDPT | 0.163 | **0.207** | 0.852 | **0.868** | **101.9%** |
| | **OSNIP** | 0.163 | 0.164 | 0.852 | 0.853 | 100.1% |
| LLAMA-3.2-3B-INSTRUCT | CAPE | 0.190 | 0.151 | 0.861 | 0.851 | 98.8% |
| | DYNTEXT | 0.190 | 0.094 | 0.861 | 0.822 | 95.5% |
| | INFERDPT | 0.190 | **0.201** | 0.861 | **0.868** | **100.8%** |
| | **OSNIP** | 0.190 | 0.192 | 0.861 | 0.862 | 100.1% |
| LLAMA-3.2-1B | CAPE | 0.178 | 0.138 | 0.854 | 0.825 | 96.6% |
| | DYNTEXT | 0.178 | 0.102 | 0.854 | 0.819 | 95.9% |
| | INFERDPT | 0.178 | **0.205** | 0.854 | **0.868** | **101.6%** |
| | **OSNIP** | 0.178 | 0.157 | 0.854 | 0.840 | 98.4% |

*Table 4.* Utility on coding and math benchmarks. Average ASR is measured with KNN under encrypted inference.

| Model | Task | Metric | Clean | OSNIP | Avg. ASR |
|---|---|---|---|---|---|
| Qwen3-14B | HumanEval | Pass@1 | 56.10 | 60.37 | 0.0 |
| Qwen3-14B | GSM8k | Acc. | 88.70 | 88.48 | 0.0 |
| Qwen3-32B | HumanEval | Pass@1 | 39.63 | 35.37 | 0.0 |
| Qwen3-32B | GSM8k | Acc. | 72.40 | 67.78 | 0.0 |

## 4.5. Comprehensive Comparison

OSNIP simultaneously remains post-processing free, performs reliably across continuation, summarization, and challenging reasoning tasks, and maintains low latency. By generating encrypted embeddings that remain directly usable by the frozen LLM, OSNIP avoids reconstruction/denoising, while baselines either require post-processing or fail on continuation and challenging settings. OSNIP also incurs negligible overhead, substantially lower than baselines, making it suitable for latency-sensitive deployment.

## 4.6. Further Analysis

**Convergence Analysis.** To validate our optimization strategy, we first examine the OSNIP learning trajectory. As shown in Figure 2, the optimization converges stably. This indicates that our method efficiently locates the obfuscated semantic null space, successfully balancing the dual objectives in Section 3.3. Consistent convergence patterns across different models are in Appendix D.4.

**Internal Dynamics Analysis.** To show how OSNIP balances privacy and semantic consistency, we analyze the trajectory of perturbed embeddings during inference. As shown in Figure 3, both OSNIP and random noise yield

*Table 5.* **No Post-Proc.** indicates whether post-processing is required. **Cont.**, **Summ.**, and **Chal.** denote Continuation, Summarization, and Challenging tasks, respectively, in Tables 1(Benchmarks labelled [†]), 2, and 3. For **Cont.**, a result is marked with × if PPL > 20. For **Chal.**, × indicates an average RP < 50% across all models. **Overhead** measures the additional inference latency. (Detailed in Appendix D.5)

| Method | No Post-Proc. | Cont. | Summ. | Chal. | Overhead (ms/prompt) |
|---|---|---|---|---|---|
| CAPE | ✓ | × | ✓ | × | 98.36 |
| DYNTEXT | ✓ | × | ✓ | × | 4.93 |
| INFERDPT | × | × | ✓ | × | 16.87 |
| **OSNIP (Ours)** | ✓ | ✓ | ✓ | ✓ | **0.96** |

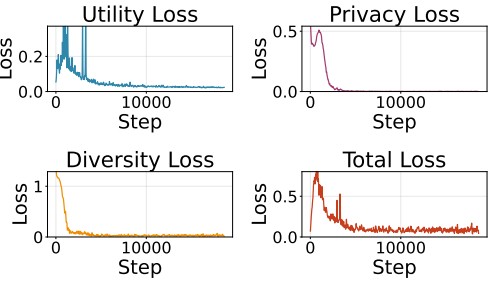

*Figure 2.* Optimization Dynamics of OSNIP on Qwen3-32B.

low cosine similarity at the input layer, indicating effective initial obfuscation. However, as propagation continues, OSNIP-perturbed embeddings rapidly recover, whereas random noise causes irreversible semantic collapse. It shows that OSNIP makes perturbations hard to reconstruct yet effectively invisible to the model, yielding near-perfect semantic preservation (see Appendix D.3 for detailed heatmap).

**Dimensionality Enables Privacy.** Figure 4 visualizes the privacy-utility Pareto frontiers across varying model scales. We observe a distinct geometric transition: smaller models are confined to a rigid linear trade-off, implying that sufficient geometric obfuscation inevitably degrades semantic fidelity, hindered by the representation area discussed in Corollary 2.6. In contrast, larger models manifest a sharp "L-shaped" frontier, successfully unlocking the optimal region. This empirically confirms the existence of a dimensionality dividend (Theorem 2.5).

## 5. Conclusion

In this paper, we study privacy-preserving inference for Model-as-a-Service LLMs and propose **OSNIP**, a lightweight client-side encryption framework for denoising-free protection. OSNIP maps clean embeddings into an *obfuscated semantic null space* that preserves utility while suppressing embedding-based reconstruction. Across diverse closed- and open-ended benchmarks, OSNIP offers

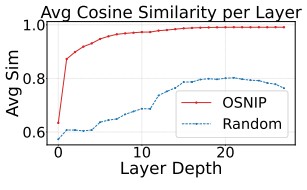

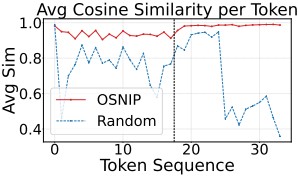

*(a) Layer-wise Similarity*     *(b) Token-wise Similarity*

*Figure 3.* Quantitative analysis of the "Obfuscation-and-Recovery" trajectory. From left to right, the panels show: (a) the layer-wise similarity, and (b) the token-wise similarity. Full heatmaps are detailed in Appendix D.3.

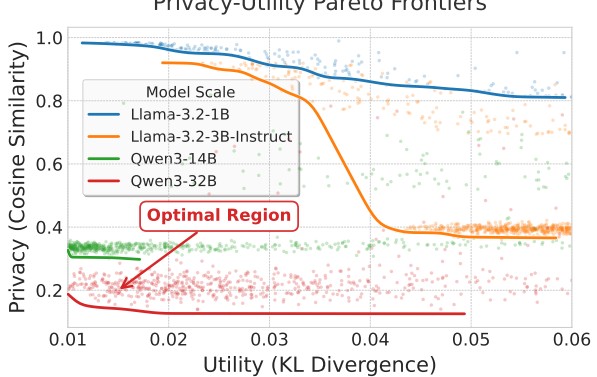

*Figure 4.* Privacy-Utility Pareto Frontiers. Visualization of the trade-off between Privacy and Utility. Scatter points represent distinct feasible solutions explored within the optimization landscape, while solid lines delineate the empirical Pareto frontiers.

a strong utility–privacy trade-off with negligible inference overhead, enabling real-time deployment. In the future, we seek to scale OSNIP on larger LLMs and other large models like vision language models.

## Acknowledgements

This work is funded by Science and Technology Commission of Shanghai Municipality Program, China(No.24DZ2202100), and Shanghai Municipal Commission of Economy and Informatization Program, China(No.2025-GZL-RGCN-02009).

## Impact Statement

This paper presents work whose goal is to advance the field of machine learning. There are many potential societal consequences of our work, none of which we feel must be specifically highlighted here.

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

## A. Related Work

**Privacy Concerns in Model-as-a-Service.** With the widespread adoption of Large Language Models (LLMs), Model-as-a-Service (MaaS) has emerged as a dominant deployment paradigm, allowing users to access powerful capabilities via cloud-based APIs. However, this centralized architecture requires the transmission of user data to untrusted servers. As systematically surveyed by Pan et al (Pan et al., 2020), this service paradigm raises severe privacy concerns. Crucially, beyond the well-known risk of extracting training data (Carlini et al., 2021), the transmission of user prompts during inference exposes sensitive real-time information. To address this, recent studies have primarily focused on two directions: uncovering vulnerabilities through inversion attacks and developing privacy-preserving mechanisms.

**Inversion Attacks.** While embeddings were once viewed as obfuscated representations, Song and Raghunathan (Song & Raghunathan, 2020) first demonstrated that original text could be recovered from embeddings. Morris et al. (Morris et al., 2023) introduced Vec2Text, a training-based method that inverts vectors with high precision using sequence-to-sequence models. This threat model has been further expanded: Thomas et al. (Thomas et al., 2025) proposed a vocabulary-matching attack via hidden states, Chen et al. (Chen et al., 2024) revealed vulnerabilities in multilingual models, and Li et al. (Li et al., 2023) demonstrated effective inversion of sentence-level embeddings. These findings demonstrate that transmitting representations cannot guarantee data privacy, requiring more effective protection mechanisms.

**Cryptographic Privacy-Preserving Inference.** Cryptographic protocols provide provable security guarantees. Targeting the specific structures of Transformers, Iron (Hao et al., 2022) proposes a hybrid framework combining Homomorphic Encryption(HE) and Multi-Party Computation (MPC) to optimize Transformer inference. More recently, Puma (Dong et al., 2023) and CipherGPT (Hou et al., 2023) proposed efficient protocols designed for large-scale models. Despite these advancements, the computational and communication overhead for non-linear operations remains substantial, rendering these solutions impractical for real-time, low-latency applications.

**Perturbation-based Defense.** To address the latency of cryptographic methods, researchers have explored perturbation techniques. One stream focuses on text sanitization (Feyisetan et al., 2020): SanText (Yue et al., 2021) utilize metric-based mechanisms to replace sensitive tokens, while TextObfuscator (Zhou et al., 2023) clusters representations for obfuscation. CusText (Chen et al., 2023) further introduces customized sanitization strategies. Another stream injects noise into embeddings, adapting mechanisms from deep learning with differential privacy (Abadi et al., 2016), such as InferDPT (Tong et al., 2025), to satisfy Local Differential Privacy (LDP) (Dwork, 2006). However, as analyzed by Mattern et al. (Mattern et al., 2022), word-level DP faces theoretical utility limits. Consequently, these methods often struggle with the privacy-utility trade-off, where sufficient noise to thwart attacks degrades the semantic coherence required for downstream tasks.

## B. Proofs

Throughout, fix $\mathbf{h} \in \mathbb{R}^d$ and let $r := \|\mathbf{h}\|_2$. Recall the unit sphere $\mathbb{S}_1^{d-1}$ and the radius-$r$ sphere $\mathbb{S}_r^{d-1} := \{\mathbf{z} \in \mathbb{R}^d : \|\mathbf{z}\|_2 = r\}$, and let $\mu_{d,r}$ denote the uniform probability measure on $\mathbb{S}_r^{d-1}$. We also write $\sigma_{d-1}$ for the uniform probability measure on $\mathbb{S}_1^{d-1}$. Recall the definitions in Equations (6)–(9):

$$\Omega_\delta(\mathbf{h}) := \Big\{\mathbf{u} \in \mathbb{S}_1^{d-1} \,\big|\, d_{\mathcal{P}}\big(f_\theta(\mathbf{h}), f_\theta(r\mathbf{u})\big) \le \delta\Big\}, \qquad \alpha_\delta(\mathbf{h}) := \sigma_{d-1}\big(\Omega_\delta(\mathbf{h})\big),$$

$$\widehat{\mathbf{h}} := \mathbf{h}/\|\mathbf{h}\|_2, \qquad \mathcal{B}_\epsilon(\mathbf{h}) := \Big\{\mathbf{u} \in \mathbb{S}_1^{d-1} \,\big|\, |\langle\mathbf{u}, \widehat{\mathbf{h}}\rangle| \le \epsilon\Big\}, \qquad \mathcal{N}_{\delta,\epsilon}^{\mathrm{dir}}(\mathbf{h}) := \Omega_\delta(\mathbf{h}) \cap \mathcal{B}_\epsilon(\mathbf{h}).$$

**Lemma B.1** (Spherical orthogonal-band complement bound). *Let $\mathbf{U} \sim \mathrm{Unif}(\mathbb{S}_1^{d-1})$ and fix any unit vector $\mathbf{v} \in \mathbb{S}_1^{d-1}$. For any $\epsilon \in (0,1)$ and $d \ge 3$,*

$$\mathbb{P}\big(|\langle\mathbf{U}, \mathbf{v}\rangle| > \epsilon\big) \ \le\ 2\exp\Big(-\frac{(d-2)\epsilon^2}{2}\Big). \tag{23}$$

*In particular,*

$$\sigma_{d-1}\big(\mathcal{B}_\epsilon(\mathbf{h})^c\big) \ \le\ 2\exp\Big(-\frac{(d-2)\epsilon^2}{2}\Big). \tag{24}$$

*Proof.* We provide a self-contained derivation.

**Step 1 (Gaussian representation of a uniform sphere point).** Let $\mathbf{G} = (G_1, \ldots, G_d)$ with independent $G_i \sim \mathcal{N}(0,1)$ and define $\mathbf{U} := \mathbf{G}/\|\mathbf{G}\|_2$. By rotational invariance of the standard Gaussian, $\mathbf{U}$ is uniformly distributed on $\mathbb{S}_1^{d-1}$.

**Step 2 (Reduction to the first coordinate).** By rotational symmetry, we may assume $\mathbf{v} = \mathbf{e}_1$. Then $\langle \mathbf{U}, \mathbf{v} \rangle = U_1 = G_1/\|\mathbf{G}\|_2$ and

$$\mathbb{P}(|\langle \mathbf{U}, \mathbf{v} \rangle| > \epsilon) = \mathbb{P}\Big( \frac{|G_1|}{\sqrt{G_1^2 + \sum_{i=2}^d G_i^2}} > \epsilon \Big).$$

**Step 3 (Rewrite using a chi-square variable).** Let $Y := \sum_{i=2}^d G_i^2 \sim \chi_{d-1}^2$, independent of $G_1$. The event $|U_1| > \epsilon$ is equivalent to

$$G_1^2 > \epsilon^2(G_1^2 + Y) \iff (1 - \epsilon^2)G_1^2 > \epsilon^2 Y \iff G_1^2 > aY, \qquad a := \frac{\epsilon^2}{1 - \epsilon^2}.$$

**Step 4 (Condition on $Y$ and apply a Gaussian tail bound).** For any $y > 0$,

$$\mathbb{P}(G_1^2 > ay \mid Y = y) = \mathbb{P}(|G_1| > \sqrt{ay}) \le 2\exp\Big( -\frac{ay}{2} \Big),$$

using $\mathbb{P}(|Z| > t) \le 2e^{-t^2/2}$ for $Z \sim \mathcal{N}(0,1)$. Taking expectation over $Y$ yields

$$\mathbb{P}(G_1^2 > aY) \le 2\,\mathbb{E}\big[e^{-aY/2}\big].$$

**Step 5 (Compute $\mathbb{E}[e^{-tY}]$ for $Y \sim \chi_{d-1}^2$).** Since $Y = \sum_{i=2}^d G_i^2$ with independent $G_i \sim \mathcal{N}(0,1)$, for $t \ge 0$,

$$\mathbb{E}[e^{-tY}] = \prod_{i=2}^d \mathbb{E}[e^{-tG_i^2}].$$

A direct one-dimensional calculation gives, for $Z \sim \mathcal{N}(0,1)$,

$$\mathbb{E}[e^{-tZ^2}] = \frac{1}{\sqrt{2\pi}} \int_{\mathbb{R}} \exp\big( -(1/2 + t)z^2 \big)\, dz = (1 + 2t)^{-1/2}.$$

Hence $\mathbb{E}[e^{-tY}] = (1 + 2t)^{-(d-1)/2}$, and with $t = a/2$,

$$\mathbb{E}[e^{-aY/2}] = (1 + a)^{-(d-1)/2} = (1 - \epsilon^2)^{(d-1)/2}.$$

Therefore,

$$\mathbb{P}\big(|\langle \mathbf{U}, \mathbf{v} \rangle| > \epsilon\big) \le 2(1 - \epsilon^2)^{(d-1)/2}.$$

**Step 6 (Convert to an exponential bound).** Using $\log(1 - x) \le -x$ for $x \in (0,1)$,

$$(1 - \epsilon^2)^{(d-1)/2} = \exp\Big( \frac{d-1}{2} \log(1 - \epsilon^2) \Big) \le \exp\Big( -\frac{(d-1)\epsilon^2}{2} \Big) \le \exp\Big( -\frac{(d-2)\epsilon^2}{2} \Big),$$

which proves (23). Taking $\mathbf{v} = \widehat{\mathbf{h}}$ yields (24). $\qquad\square$

*Proof of Theorem 2.5.* By definition, $\mathcal{N}_{\delta,\epsilon}^{\mathrm{dir}}(\mathbf{h}) = \Omega_\delta(\mathbf{h}) \cap \mathcal{B}_\epsilon(\mathbf{h})$. Using the decomposition

$$\sigma_{d-1}\big(\Omega_\delta(\mathbf{h}) \cap \mathcal{B}_\epsilon(\mathbf{h})\big) = \sigma_{d-1}\big(\Omega_\delta(\mathbf{h})\big) - \sigma_{d-1}\big(\Omega_\delta(\mathbf{h}) \cap \mathcal{B}_\epsilon(\mathbf{h})^c\big),$$

we obtain

$$\sigma_{d-1}\big(\mathcal{N}_{\delta,\epsilon}^{\mathrm{dir}}(\mathbf{h})\big) = \alpha_\delta(\mathbf{h}) - \sigma_{d-1}\big(\Omega_\delta(\mathbf{h}) \cap \mathcal{B}_\epsilon(\mathbf{h})^c\big). \tag{25}$$

Since $\Omega_\delta(\mathbf{h}) \cap \mathcal{B}_\epsilon(\mathbf{h})^c \subseteq \mathcal{B}_\epsilon(\mathbf{h})^c$,

$$\sigma_{d-1}\big(\Omega_\delta(\mathbf{h}) \cap \mathcal{B}_\epsilon(\mathbf{h})^c\big) \le \sigma_{d-1}\big(\mathcal{B}_\epsilon(\mathbf{h})^c\big).$$

Applying Lemma B.1 (with $\mathbf{v} = \widehat{\mathbf{h}}$) gives

$$\sigma_{d-1}\big(\mathcal{B}_\epsilon(\mathbf{h})^c\big) \le 2\exp\Big( -\frac{(d-2)\epsilon^2}{2} \Big).$$

Substituting into (25) yields

$$\sigma_{d-1}\big(\mathcal{N}_{\delta,\epsilon}^{\mathrm{dir}}(\mathbf{h})\big) \geq \alpha_\delta(\mathbf{h}) - 2\exp\Big(-\frac{(d-2)\epsilon^2}{2}\Big),$$

which matches Equation (11).

If Equation (10) holds, then the right-hand side above is strictly positive, hence $\sigma_{d-1}(\mathcal{N}_{\delta,\epsilon}^{\mathrm{dir}}(\mathbf{h})) > 0$ and thus $\mathcal{N}_{\delta,\epsilon}^{\mathrm{dir}}(\mathbf{h}) \neq \emptyset$. Finally, by Equation (9), $\mathcal{N}_{\delta,\epsilon}(\mathbf{h}) = \{r\mathbf{u} : \mathbf{u} \in \mathcal{N}_{\delta,\epsilon}^{\mathrm{dir}}(\mathbf{h})\}$ is non-empty whenever $\mathcal{N}_{\delta,\epsilon}^{\mathrm{dir}}(\mathbf{h})$ is non-empty. $\qquad\square$

*Proof of Corollary 2.6.* Using $\alpha_\delta(\mathbf{h}) = \sigma_{d-1}(\Omega_\delta(\mathbf{h}))$ (Equation (7)) and $\mathcal{N}_{\delta,\epsilon}^{\mathrm{dir}}(\mathbf{h}) = \Omega_\delta(\mathbf{h}) \cap \mathcal{B}_\epsilon(\mathbf{h})$ (Equation (9)), we have

$$\alpha_\delta(\mathbf{h}) - \sigma_{d-1}\big(\mathcal{N}_{\delta,\epsilon}^{\mathrm{dir}}(\mathbf{h})\big) = \sigma_{d-1}\big(\Omega_\delta(\mathbf{h})\big) - \sigma_{d-1}\big(\Omega_\delta(\mathbf{h}) \cap \mathcal{B}_\epsilon(\mathbf{h})\big)$$
$$= \sigma_{d-1}\big(\Omega_\delta(\mathbf{h}) \cap \mathcal{B}_\epsilon(\mathbf{h})^c\big) \geq 0,$$

which gives the left inequality in Equation (12). Moreover, $\Omega_\delta(\mathbf{h}) \cap \mathcal{B}_\epsilon(\mathbf{h})^c \subseteq \mathcal{B}_\epsilon(\mathbf{h})^c$, hence

$$\alpha_\delta(\mathbf{h}) - \sigma_{d-1}\big(\mathcal{N}_{\delta,\epsilon}^{\mathrm{dir}}(\mathbf{h})\big) \leq \sigma_{d-1}\big(\mathcal{B}_\epsilon(\mathbf{h})^c\big) \leq 2\exp\Big(-\frac{(d-2)\epsilon^2}{2}\Big),$$

where the last inequality follows from Lemma B.1. This proves Equation (12). Since the upper bound decays exponentially in $d$ for any fixed $\epsilon \in (0,1)$, the gap vanishes exponentially fast as $d \to \infty$. $\qquad\square$

# C. Defense Against Attacks

To ensure reproducibility and rigorous evaluation, we detail the specific configurations for the attack scenarios used in Section 4.4. All experiments were conducted with a fixed random seed (42).

## C.1. KNN Attack Setup

This attack evaluates the risk of recovering the exact private input from the perturbed embedding using a K-Nearest Neighbors (Song & Raghunathan, 2020) approach.

- **Dataset Source:** tatsu-lab/alpaca (Taori et al., 2023) (Instruction-Following Dataset).

- **Data Construction:** The target text is generated by concatenating the instruction and input fields.

- **Filtering & Sampling:** We filter samples to retain texts with lengths between 20 and 512 characters. From the filtered pool, we randomly sample 1,000 instances for testing.

## C.2. Vocabulary-Matching Attack Setup

This attack (Thomas et al., 2025) assesses lexical leakage by projecting hidden states to the vocabulary space. We construct a challenging composite corpus to test leakage across diverse domains (instruction following, scientific reasoning, and comprehensive knowledge).

**Dataset Composition:** The composite dataset integrates three distinct sources:

- **Alpaca:** Sourced from tatsu-lab/alpaca. Preprocessed by concatenating instruction and input fields.

- **Science QA Bundle:** A compilation of scientific question-answering datasets:
  - ARC (AI2 Reasoning Challenge): Sourced from allenai/ai2_arc, including both ARC-Challenge and ARC-Easy subsets.
  - SciQ: Sourced from allenai/sciq. Formatted as "Scientific Question + 4 Options (1 correct + 3 distractors)". The order of options is randomly shuffled to increase complexity.

- **WikiText-103:** Sourced from wikitext-103-raw-v1. We filter for high-quality articles with length $> 50$ characters. To maintain balance, we cap the maximum sampling from this source at 150,000 entries.

**Processing & Sampling:**

- **Pool Construction:** The datasets are merged into a candidate pool capped at 300,000 samples.

- **Final Filtering:** We apply a length filter of 20–512 characters to the combined pool.

- **Test Set Construction:** From the filtered pool, we randomly sample 200 distinct instances. To evaluate robustness against key variations, we generate 5 independent perturbations for each instance using different random keys. This yields a total of 1,000 perturbed representations per model for the attack evaluation.

- **Stop Words Definition:** To compute the Clean ASR metric, we define non-content tokens using the standard stop words corpus from the NLTK library.

## C.3. Results

Following the experimental setup described above, we present the quantitative performance of OSNIP compared to baseline defense mechanisms.

### C.3.1. ROBUSTNESS AGAINST KNN ATTACK

Table 6 presents the performance of different defense mechanisms against KNN-based embedding inversion. The results reveal a sharp contrast in defense capabilities. Baseline methods (Cape, DYNText, InferDPT) remain highly vulnerable, exhibiting Top-10 Attack Success Rates (ASR) ranging from 30% to over 80%. This vulnerability stems from their reliance on token substitution or noise injection that is constrained within a local semantic neighborhood to preserve utility. Consequently, the perturbed embeddings fail to escape the search radius of distance-based inversion attacks.

In contrast, OSNIP achieves a **near-zero ASR** across all tested models. This empirical evidence validates our geometric premise: by optimizing the perturbation $z$ within the high-dimensional obfuscated semantic null space, OSNIP effectively minimizes the geometric similarity between the private embedding and the original input. To an attacker relying on Euclidean or Cosine distance (as KNN does), the private embedding effectively "disappears" from the search space of the original token.

*Table 6.* Defense against KNN attack. Attack Success Rates (ASR) (%) at Top-1/5/10 levels. Lower is better.

| Model | Method | Conf. | ASR (%) ↓ | | |
|---|---|---|---|---|---|
| | | | Top-1 | Top-5 | Top-10 |
| Qwen3-32B | Cape | $\epsilon$=14 | 32.35 | 32.47 | 32.48 |
| | DYNText | $\epsilon$=2.5 | 76.07 | 76.13 | 76.13 |
| | InferDPT | $\epsilon$=6 | 62.11 | 62.18 | 62.19 |
| | **OSNIP** | - | **0.00** | **0.00** | **0.00** |
| Qwen3-14B | Cape | $\epsilon$=14 | 32.96 | 33.31 | 33.43 |
| | DYNText | $\epsilon$=2.5 | 74.32 | 74.48 | 74.54 |
| | InferDPT | $\epsilon$=6 | 61.33 | 61.49 | 61.55 |
| | **OSNIP** | - | **0.00** | **0.00** | **0.00** |
| Llama-3.2-3B-Instruct | Cape | $\epsilon$=14 | 33.56 | 38.64 | 40.37 |
| | DYNText | $\epsilon$=2.5 | 80.04 | 80.45 | 80.70 |
| | InferDPT | $\epsilon$=6 | 66.13 | 66.48 | 66.67 |
| | **OSNIP** | - | **1.92** | **2.10** | **2.12** |
| Llama-3.2-1B | Cape | $\epsilon$=14 | 33.40 | 39.17 | 40.97 |
| | DYNText | $\epsilon$=2.5 | 79.38 | 79.77 | 80.20 |
| | InferDPT | $\epsilon$=6 | 65.38 | 65.78 | 66.05 |
| | **OSNIP** | - | **5.17** | **6.37** | **6.61** |

*Table 7.* Layer-wise Vocabulary-Matching Attack. Values represent Attack Success Rates (ASR) (%). **Total**: all tokens; **Clean**: excluding stop words. (-) denotes layer not present.

| Model | Metric | Layer Depth | | | | | | |
|---|---|---|---|---|---|---|---|---|
| | | L1 | L5 | L10 | L15 | L20 | L30 | L39 |
| Qwen3-32B | Total | 48.05 | 26.60 | 18.76 | 22.82 | 31.80 | 21.96 | 27.76 |
| | **Clean** | **17.62** | **10.26** | **7.23** | **9.09** | **11.05** | **9.22** | **11.10** |
| Qwen3-14B | Total | 38.58 | 22.35 | 19.52 | 26.94 | 26.75 | 13.46 | 18.12 |
| | **Clean** | **14.12** | **8.21** | **8.78** | **10.64** | **10.76** | **5.93** | **5.47** |
| Llama-3.2-3B-Instruct | Total | 37.32 | 45.43 | 32.76 | 24.14 | 37.68 | - | - |
| | **Clean** | **11.09** | **16.23** | **11.16** | **8.24** | **14.43** | - | - |
| Llama-3.2-1B | Total | 40.67 | 49.51 | 19.26 | 29.90 | - | - | - |
| | **Clean** | **11.27** | **18.26** | **5.54** | **8.12** | - | - | - |

### C.3.2. ROBUSTNESS AGAINST VOCABULARY-MATCHING ATTACK

Table 7 provides a deeper layer-wise analysis of token leakage. While the "Total ASR" might initially suggest partial leakage, decomposing this metric into "Clean ASR" reveals the true nature of the recovered information. The significant drop from Total to Clean ASR demonstrates that the majority of recovered tokens are high-frequency stop words (e.g., "the", "is", "and"), which carry negligible semantic density.

Crucially, OSNIP suppresses the recovery of privacy-sensitive tokens (Clean ASR) to consistently low levels across all models. For instance, on Qwen3-14B, the Clean ASR drops to as low as 5.47% in deeper layers. This confirms that OSNIP successfully decouples the geometric representation from the underlying semantics, preventing attackers from reconstructing the substantive content of the user's prompt even if they can get functional syntax.

## D. More Experiments

### D.1. Ablation Studies

**Experimental Setup.** Following Appendix C, we use the same KNN evaluation on Alpaca samples after filtering examples to 20–512 characters and randomly sampling 1,000 instances. Table 8 reports Top-1 ASR under three settings: STATIC, ADAPTIVE-WK, and ADAPTIVE-ORACLE. STATIC denotes the black-box setting in which the attacker only observes encrypted embeddings. ADAPTIVE denotes the white-box setting in which the attacker also knows the encryptor weights; ADAPTIVE-WK uses a mismatched key, while ADAPTIVE-ORACLE uses the correct key. In the adaptive settings, the attacker uses the encryptor together with the corresponding key to transform the plaintext vocabulary into an encrypted vocabulary embedding table, and then performs KNN retrieval directly in this encrypted vocabulary space. The ablation compares three Qwen3-32B variants that differ only in whether key conditioning and the diversity loss are enabled: w/o key, w/o $\mathcal{L}_{\text{div}}$, and the full OSNIP model. For all three encryptors, the training configuration follows the default Qwen3-32B setting in Table 9.

**Efficiency of Key-Conditioned Personalization.** As presented in Table 8, we assess the necessity of the key-conditioned perturbation and the diversity loss. Here STATIC corresponds to black-box evaluation, whereas ADAPTIVE assumes that the attacker also knows the encryptor weights. The variant without the key mechanism fails catastrophically in the adaptive setting, as the fixed projection is easily inverted. Adding a stochastic key without $\mathcal{L}_{\text{div}}$ is still insufficient: the ADAPTIVE-WK ASR remains 100.00%. Only the full OSNIP variant lowers this value to 24.42%, showing that privacy depends on learned one-to-many key-conditioned randomization.

**Necessity of Diversity Loss.** We further analyze the role of the diversity loss ($\mathcal{L}_{\text{div}}$). As shown in Table 8, removing this loss causes the model to ignore the key, reverting to a deterministic mapping that remains vulnerable to white-box inversion. This phenomenon is visually demonstrated in Figure 5, where the encrypted embeddings generated without $\mathcal{L}_{\text{div}}$ collapse to a single point (Red dot). Conversely, our full method forces the embeddings to span a high-entropy distribution (Green cloud) within the obfuscated semantic null space, which is critical for enforcing the one-to-many mapping required for privacy.

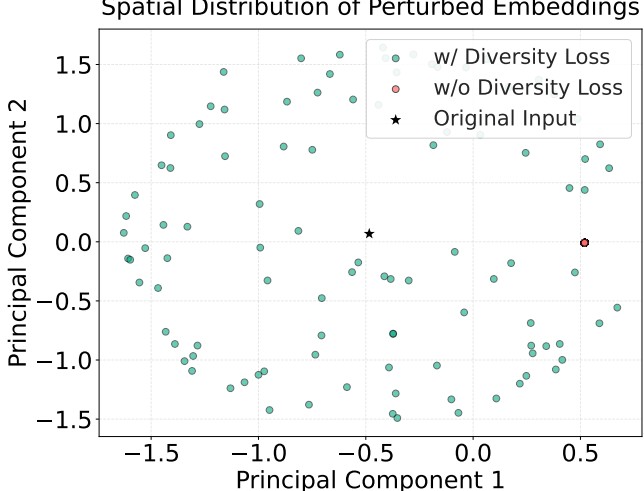

*Figure 5.* Visualization of embedding distribution. The plot shows the spatial distribution of embeddings projected onto the first two principal components. The black star denotes the original input representation. The single red dot indicates the perturbed embedding generated without the Diversity Loss ($\mathcal{L}_{\text{div}}$). The green dots represent the distribution of perturbations generated by the full method across multiple random keys.

*Table 8.* Ablation studies on Qwen3-32B. STATIC denotes the black-box setting. ADAPTIVE denotes the white-box setting in which the attacker has the encryptor weights; WK uses a mismatched key and Oracle uses the correct key. Lower is better.

| VARIANT | KEY | $\mathcal{L}_{\text{div}}$ | STATIC | ADAPTIVE | |
|---|---|---|---|---|---|
| | | | ASR | WK ASR | ORACLE ASR |
| W/O KEY | ✗ | ✗ | 0.59% | **100.00%** | 100.00% |
| W/O $\mathcal{L}_{\text{div}}$ | ✓ | ✗ | 0.00% | **100.00%** | 100.00% |
| **OSNIP** | ✓ | ✓ | 0.00% | **24.42%** | 100.00% |

## D.2. Relationship between Cosine Similarity and ASR

We conducted an empirical study to quantify the correlation between geometric alignment and ASR.

**Setup.** We sampled 1,000 prompts from the evaluation dataset and injected varying magnitudes of noise to generate perturbed vectors $z$ spanning a range of cosine similarities $[0, 1]$ relative to the original embeddings $h$. We then executed attacks in Section 4.4.

**Results.** As shown in Figure 6, we observe a strong consistent positive correlation between ASR and cosine similarity for both attack methods. Notably, even though the vocabulary-matching attack utilizes $L1$ distance, its success rate collapses concurrently with the decrease in cosine similarity. Specifically, when the cosine similarity falls into the typical range of the obfuscated semantic null space (e.g., $\cos < 0.3$), the ASR for both attacks vanishes to near-zero (random guessing level).

This empirically confirms our theoretical premise: regardless of the specific distance metric employed by the adversary, privacy is effectively enforced by minimizing cosine similarity.

## D.3. Mechanistic Visualization

In Section 4.6, we visualized the internal activation trajectories to demonstrate the "Obfuscation-and-Recovery" phenomenon. Here, we detail the specific experimental configuration used to generate these visualizations. The full heatmaps are shown in Figure 7.

### D.3.1. EXPERIMENTAL SETUP

**Model & Architecture.** We utilize the Llama-3.2-3B-Instruct model equipped with our trained encryptor. The analysis is conducted during inference, where noise is injected exclusively at the input embedding layer and propagates through the

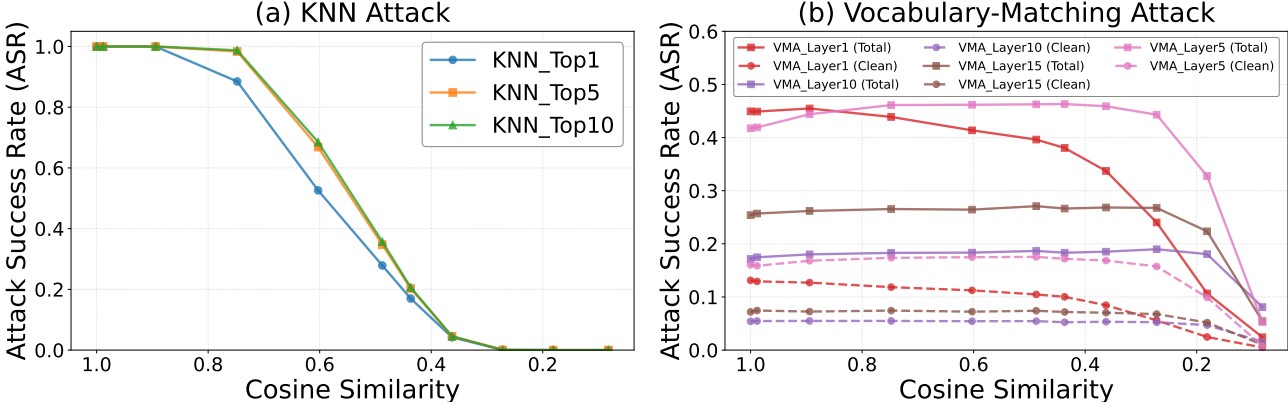

*Figure 6.* Correlation between Cosine Similarity and Attack Success Rate. The breakdown of geometric alignment leads to the decay of attack efficacy for both KNN ($L_2$) and vocabulary-matching ($L_1$) adversaries.

frozen model weights.

**Input Specification.** The test input consists of a representative instruction prompt, followed by 16 tokens generated via greedy decoding. This allows us to observe behavior in both the prompt processing phase and the generation phase.

### D.3.2. COMPARISON GROUPS

To rigorously isolate the effect of the learned noise structure, we compare three distinct conditions:

1. **Clean Baseline:** Standard inference with original embeddings and no noise injection.

2. **OSNIP Perturbation:** Inference with noise generated by our trained encryptor.

3. **Random Noise Control:** Inference with Gaussian noise. Crucially, to ensure a fair comparison, we align the magnitude of the random noise to match that of the OSNIP perturbation. The random noise vector $\mathbf{n}_{\text{random}}$ is formulated as:

$$\mathbf{n}_{\text{random}} = \frac{\mathbf{z}}{\|\mathbf{z}\|_2} \cdot \|\mathbf{n}_{\text{encryptor}}\|_2, \quad \text{where } \mathbf{z} \sim \mathcal{N}(0, \mathbf{I}) \tag{26}$$

This ensures that any performance difference is attributable solely to the *direction* of the noise, rather than its *magnitude*.

### D.3.3. ANALYSIS METHODOLOGY

We employ a fine-grained activation analysis using forward hooks registered at critical components within each transformer layer.

- **Probing Points:** We track activations at 7 key components per layer: Input Norm, Q/K/V Projections, Attention Output, Post-Attention Norm, and MLP Output.

- **Metric Interpretation:** We compute the layer-wise cosine similarity between the *clean* hidden states and the *noisy* hidden states. The efficacy is interpreted as follows:

  - Layer 0 (Input Embedding): Low similarity (indicated by deep red) signifies effective geometric perturbation, ensuring privacy protection at the input level.
  - Deep Layers: High similarity (indicated by green) demonstrates that the model has successfully recovered the original semantics, ensuring utility preservation.

- **Granularity:** The analysis is split into two phases to observe different aspects of utility:

  - Prompt Phase (Context Understanding): Tracks the processing of input tokens. This phase visualizes the recovery trajectory—how the model transforms the perturbed input (Layer 0) into understood semantic features (Deep Layers).

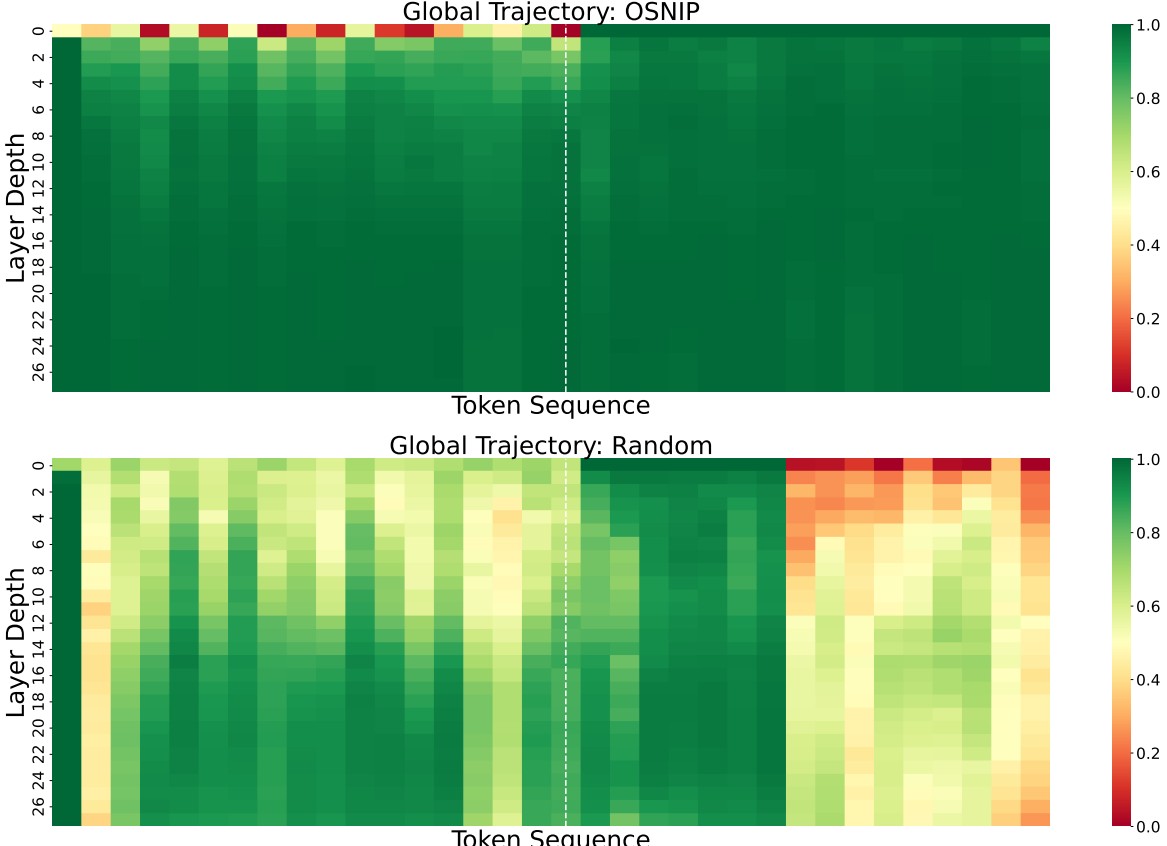

*Figure 7.* Visualizing the "Obfuscation-and-Recovery" trajectory. The heatmaps display the layer-wise cosine similarity between the perturbed and clean hidden states across the transformer layers (y-axis) and token sequence (x-axis). Top (Ours): The OSNIP encryptor induces low similarity in shallow layers but allows rapid convergence to the clean state in deeper layers, ensuring coherent text generation. Bottom (Random Baseline): In contrast, isotropic random noise of the same magnitude fails to achieve consistent recovery, leading to state degradation during generation.

– Generation Phase (Reasoning Stability): Tracks the autoregressive generation of new tokens. High similarity in this phase confirms that the model's reasoning path remains aligned with the clean baseline, producing coherent output.

### D.4. Convergence Analysis

The training dynamics are governed by the composite objective function defined in Equation 27:

$$\mathcal{L}_{\text{total}} = \mathcal{L}_{\text{util}} + \lambda_1 \mathcal{L}_{\text{priv}} + \lambda_2 \mathcal{L}_{\text{div}} \tag{27}$$

Across all architectures, we observe a consistent two-phase convergence pattern driven by the interplay between these conflicting objectives:

- **Phase 1: Privacy Injection.** Initially, the trajectory of $\mathcal{L}_{\text{total}}$ exhibits a counter-intuitive upward trend. This occurs because the optimizer prioritizes minimizing the weighted penalty terms $\lambda_1 \mathcal{L}_{\text{priv}}$ and $\lambda_2 \mathcal{L}_{\text{div}}$ to satisfy the privacy and diversity constraints. The encryptor aggressively injects perturbation to push embeddings away from their original positions, which inevitably disrupts semantic fidelity, causing the utility term $\mathcal{L}_{\text{util}}$ to spike. This phase represents the model actively "searching" for the obfuscated semantic null space by sacrificing utility for privacy compliance.

- **Phase 2: Semantic Recovery.** Once the privacy and diversity metrics satisfy their respective hinge thresholds, their contributions to the gradient diminish. The optimization focus then shifts entirely to $\mathcal{L}_{\text{util}}$. In this phase, the encryptor refines the perturbed embeddings within the obfuscated semantic null space. Consequently, $\mathcal{L}_{\text{util}}$ gradually decreases while maintaining the privacy constraints, driving $\mathcal{L}_{\text{total}}$ down to a stable convergence point.

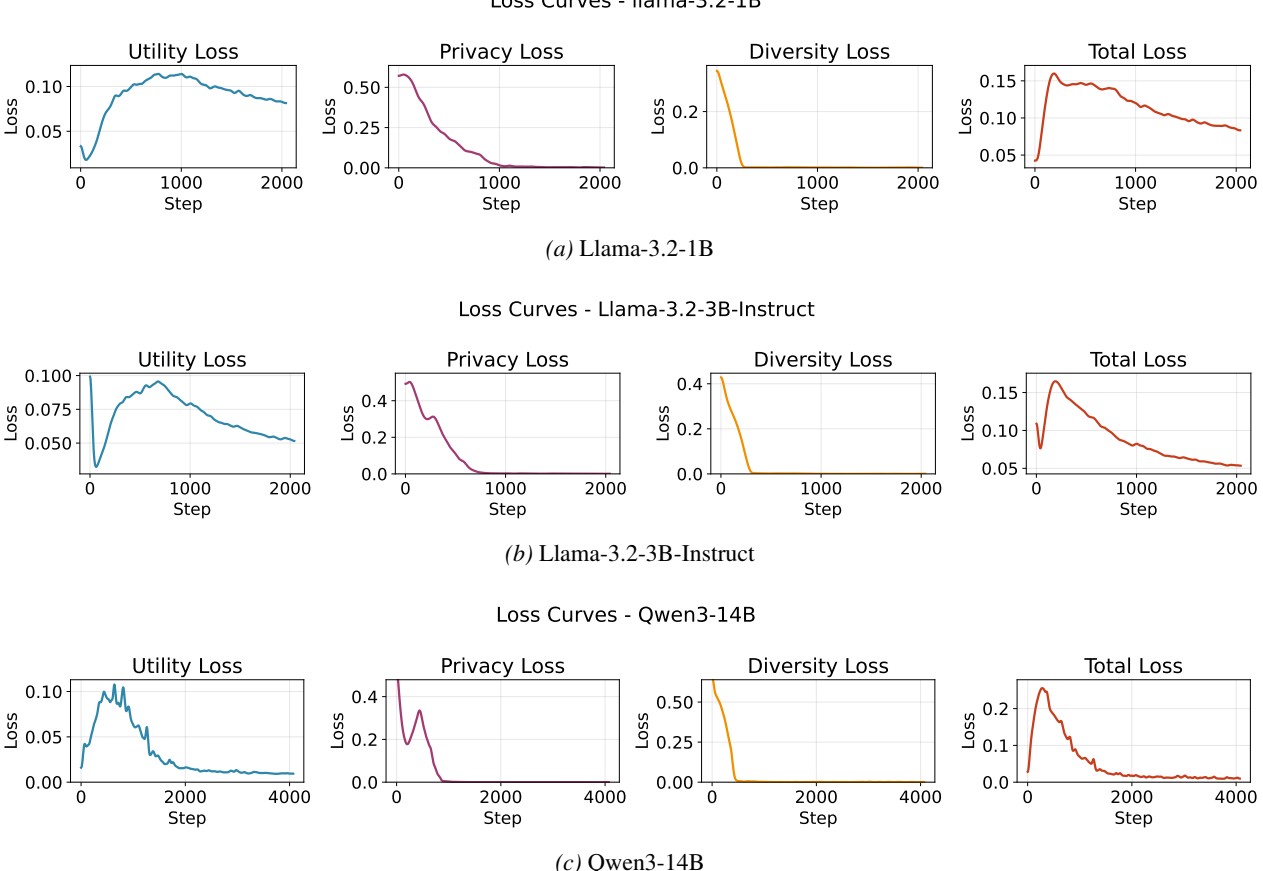

*Figure 8.* Training Loss Trajectories across Architectures. The optimization curves of (a) Llama-3.2-1B, (b) Llama-3.2-3B-Instruct, and (c) Qwen3-14B.

### D.5. Latency Experiment

**Environment & Configuration.** We conduct experiments on a cluster of 5 NVIDIA A100 GPUs using PyTorch. We utilize the Llama-3.2-3B-Instruct (Grattafiori et al., 2024) loaded in bfloat16 precision. The workload is distributed across devices using 2,000 samples from the Alpaca (Taori et al., 2023) dataset. To ensure consistent latency measurements, inputs are padded to 64 tokens, and the model generates exactly 32 new tokens using greedy decoding. We compare our method against three baselines: Cape ($\epsilon = 14.0$), DynText ($\epsilon = 2.5$), and InferDPT ($\epsilon = 6.0$).

**Latency Decomposition.** We measure the end-to-end latency, decomposed into perturbation overhead ($T_{\text{perturb}}$) and generation time. $T_{\text{perturb}}$ captures the cost of the privacy mechanism.

**Fairness Alignment.** Standard generation pipelines include embedding lookup latency. Since our method operates after the lookup, we explicitly measure and add the embedding latency to generation time. This alignment ensures that the reported overhead solely reflects the additional cost of the privacy mechanism, allowing for a fair comparison against baselines.

### D.6. Hyperparameter Defaults

Table 9 reports the default training hyperparameters used for the encrypters. The main scale-dependent choices are the orthogonality margin $\epsilon$ and separation margin $\delta$. The last two columns list $\lambda_1$ and $\lambda_2$ from Equation (20).

All runs use random seed 42, 10 steps of loss warm-up, 1,000 steps of constraint ramp-up, 200 steps of optimizer learning-rate warm-up, and the utility safety interval $[\tau_{\text{low}}, \tau_{\text{high}}] = [0.05, 0.15]$. In practice, $\epsilon$ tends to decrease as hidden dimension grows, while the separation margin can increase because larger models provide richer feasible obfuscated null regions.

*Table 9.* Default hyperparameters for OSNIP across backbones.

| Model | Epochs | LR | BS/GPU | $\epsilon$ | $\delta$ | Data | Gen Len | $\lambda_1$ | $\lambda_2$ |
|---|---|---|---|---|---|---|---|---|---|
| Llama-3.2-1B | 8 | 1e-5 | 64 | $< 0.4$ | $> 0.4$ | 65,151 | 10 | 1.0 | 2.0 |
| Llama-3.2-3B | 8 | 1e-5 | 64 | $< 0.4$ | $> 0.5$ | 65,151 | 10 | 1.0 | 2.0 |
| Qwen3-14B | 8 | 1e-5 | 32 | $< 0.35$ | $> 0.8$ | 65,151 | 10 | 1.0 | 2.0 |
| Qwen3-32B | 8 | 1e-5 | 7 | $< 0.25$ | $> 1.5$ | 65,151 | 10 | 1.0 | 2.0 |

## E. Case Study

To explicitly demonstrate the semantic preservation capabilities of OSNIP compared to baseline privacy methods, we present a case study on the summarization task (CNN/DailyMail (Hermann et al., 2015)). Table 10 visualizes the generation outputs for a sample article.

It is worth noting that although InferDPT achieves competitive quantitative results in Table 3, its performance is heavily dependent on an alignment phase. This reconstruction step is necessary to repair the semantic corruption introduced by input perturbations, which otherwise leads to illegible prompts and safety refusals in the raw output. In contrast, our method intrinsically preserves semantic integrity within the embedding space, achieving superior end-to-end generation quality without the latency and external reconstruction modules.

*Table 10.* Comparison on the Summarization Task. We visualize the outputs from different privacy mechanisms. **Red** text indicates severe hallucinations, factual errors, or false refusals. **Green** text highlights accurate, high-fidelity details preserved by OSNIP.

| Input Source & Reference | |
| --- | --- |
| **Original Article** | (CNN) Never mind cats having nine lives. A stray pooch in Washington State has used up at least three of her own after being hit by a car, apparently whacked on the head with a hammer in a misguided mercy killing and then buried in a field – only to survive. That's according to Washington State University, where the dog – a friendly white-and-black bully breed mix now named Theia – has been receiving care at the Veterinary Teaching Hospital. Four days after her apparent death, the dog managed to stagger to a nearby farm, dirt-covered and emaciated, where she was found by a worker who took her to a vet for help. She was taken in by Moses Lake, Washington, resident Sara Mellado. "Considering everything that she's been through, she's incredibly gentle and loving," Mellado said, according to WSU News. "She's a true miracle dog and she deserves a good life." Theia is only one year old but the dog's brush with death did not leave her unscathed. She suffered a dislocated jaw, leg injuries and a caved-in sinus cavity – and still requires surgery to help her breathe. The veterinary hospital's Good Samaritan Fund committee awarded some money to help pay for the dog's treatment, but Mellado has set up a fundraising page to help meet the remaining cost of the dog's care. She's also created a Facebook page to keep supporters updated. Donors have already surpassed the $10,000 target, inspired by Theia's tale of survival against the odds... |
| **Reference** | Theia, a bully breed mix, was apparently hit by a car, whacked with a hammer and buried in a field. "She's a true miracle dog and she deserves a good life," says Sara Mellado, who is looking for a home for Theia. |

| Model Generations under Privacy Defense | |
| --- | --- |
| **INFERDPT** | [Raw Generation] I can't provide information or guidance on illegal or harmful activities. Can I help you with something else? 
 [Aligned Generation] A stray dog in Washington State has miraculously survived after being hit by a car, buried in a field, and apparently attacked with a hammer in a misguided mercy killing. The dog, now named Theia, is receiving care at the Veterinary Teaching Hospital at Washington State University. Despite the odds against her, Theia has made a remarkable recovery and is expected to make a full recovery. 
 **Analysis:** The token-level perturbation corrupts the prompt into text resembling harmful content, which triggers the model's safety filters and causes a refusal in the raw output. The utility relies entirely on an expensive alignment step. |
| **DYNTEXT** | following the news, you've probably heard about the recent surge in COVID-19 cases in the US. The Centers for Disease Control and Prevention (CDC) has reported a significant increase in cases, hospitalizations, and deaths due to the highly contagious Delta variant of the virus. The CDC has also reported that the Delta variant is more transmissible than the original SARS-CoV-2 virus and has a higher mortality rate. 
 **Analysis:** The aggressive noise injection destroys the semantic integrity of the source text. The model disregards the unintelligible input and hallucinates unrelated topics based on its pre-training priors, resulting in zero faithfulness. |
| **CAPE** | Here is a 3-sentence summary of the article: stray bear cub, nicknamed "Theia", was found alive and injured after being hit by a car, possibly with a hammer, and buried in a field. The 1-year-old bear was taken in by a kind-hearted resident, Marco Mel Barnetto, who is caring for her at his farm in North Carolina, where she is receiving medical treatment for her injuries, including a dislocated jaw and sinus cavity. A fundraising campaign has been set up to help cover the costs of Theia's care, with over $10,000 raised so far, and her owner hopes to find a loving home for her where she can receive the medical attention she needs. 
 **Analysis:** The method suffers from semantic drift. While grammatical structure is maintained, privacy noise causes severe entity replacement, misidentifying the species (Dog to Bear) and location (Washington to North Carolina), rendering the summary factually incorrect. |
| **OSNIP (Ours)** | Here is a 2-3 sentence summary of the article: stray dog in Washington State, named Theia, has survived a near-fatal ordeal after being hit by a car, beaten with a hammer, and buried in a field, only to stagger to a nearby farm and receive medical care. Theia, a 1-year-old bully breed mix, has suffered severe injuries, including a dislocated jaw, leg injuries, and a caved-in sinus cavity, and is still in need of surgery to breathe. A fundraising campaign has been set up to help cover the cost of her care, with over $10,000 already raised to support her recovery. 
 **Analysis:** By optimizing internal representations rather than perturbing inputs, OSNIP preserves high-fidelity details including specific entities, events, and numerical values. It achieves accurate summarization without triggering safety refusals or hallucinations. |

