# OpenReview forum: "OSNIP: Balancing the Privacy-Utility-Efficiency Trilemma in LLM Inference via Obfuscated Semantic Null Space"
_ICML.cc/2026/Conference — ICML 2026 regular_

### Official Review · Reviewer_nqEZ · 2026-03-09

**Soundness:** 3
**Presentation:** 3
**Significance:** 2
**Originality:** 3
**Overall Recommendation:** 5
**Confidence:** 4

**Summary:**

The paper discusses privacy-preserve LLM inference under MaaS setting. The question to explore is whether one can get privacy and meanwhile high utility and low inference bottleneck by embedding perturbation into "obfuscated semantic null space". With its definition, the paper proposes OSNIP, the client side encryptor with good trade-offs. The empirical study show that OSNIP preserves utility with acceptable latency overhead and achieve KNN leakage close to zero.

**Compliance With Llm Reviewing Policy:**

Affirmed.

**Final Justification:**

My questions are solved and I think its a good paper.

**Key Questions For Authors:**

1. The theorem's assumption is based on t he semantic coverage rate. Is there any method to empirically quantify this, such as MC, or any proxy to illustrate it?

2. Please state clearly if: OSNIP is defined as a formal privacy mechanism, or it's an empirical defense against a restricted family of embedding inversion attacks.

3. About the 3-rd party, I have a concern. If the model is updated, fine-tuned, or changed, how to refresh the encryptor?

I would consider raising my score if the questions are well-addressed.

**Limitations:**

Please refer to questions and weaknesses.

**Strengths And Weaknesses:**

Strength:

1. The novel high-level idea. The contribution is to reframe high-dimension as a helpful privacy helper. This seems to be a fresh angle compared with existing frames.

2. The good problem formulation.  The system model, and adversary models are easy to follow. And OSNIP is defined in a clean manner. To preserve predictive behavior with KL, and directed decor relation with cosine hinge loss, and stabilize training with utility-gaware curriculum.

Weakness:

1. The theory requires an unmeasured quantity. Theorem 2.5 conditions on semantic coverage rate, which is not estimated or bounded or empirically probed by the paper with real-world models. Thereby, although the theorem is tidy, it seems to be operationally incomplete.

2. The deployment requires a trusted third party with gradient access to the server LLMs. This is essential for training encryptor. This is a non-trivial assumption (Please correct me if I'm wrong here).

3. Occasional overclaiming: breaking the trilemma seems to be absolute for a method that is validated on a specific adversary class and four open-source models.[

---

> ### Author Rebuttal · Authors · 2026-03-31
>
> We thank the reviewer for the constructive feedback and willingness to reconsider the score.
>
> >W1, Q1: Semantic coverage rate
>
> We agree the paper doesn't directly estimate the semantic coverage rate for real models; we will make this limitation explicit.
>
> The role of Theorem 2.5 is therefore not to serve as an operational certification criterion, but as a sufficient geometric existence result: if the semantic-preserving set has non-negligible coverage, then its intersection with the $\epsilon$-orthogonal band is guaranteed to be non-empty. By Lem B.1, the relevant threshold is $2\exp(-\frac{(d-2)\epsilon^2}{2})$, which decays exponentially with the representation dimension $d$. For example, if $d=4096$ and $\epsilon=0.05$, this threshold is about $0.012$, and even at $d=2048$ it is about $0.155$; these are illustrative thresholds rather than empirical estimates of $\alpha_\delta(h)$. Accordingly, our empirical observation that the privacy-utility tension is alleviated from 1B to 32B is consistent with this geometric mechanism, but does not itself measure semantic coverage.
>
> A direct empirical probe would require separate model-specific directional sampling over the iso-norm sphere, which we do not report in the current submission. In the revision, we will add a short numerical discussion in the appendix so that Theorem 2.5 is presented as a sufficient existence result rather than a directly measured operational guarantee.
>
> >W2: TTP with gradient access
>
> This paper proposes a privacy-preserving solution in which the encryptor is trained by a TTP when the model is small or the client is computationally constrained. When the model is sufficiently large, experiments indicate that the MaaS provider can train the encryptor once and distribute it to clients, while protection at inference time remains on the client side through secret keys. We validate this with white-box wrong-key (WK) attacks—the attacker has full encryptor weights but not the correct key—under both KNN and Vocabulary-Matching Attack (VMA) settings (RP on arc_easy + piqa + mnli; Qwen3-32B VMA pending):
>
> |Model|key_l2|RP(%)|BB ASR(%)|WK ASR(%)(KNN, VMA)|
> |-|-|-|-|-|
> |Llama-3.2-1B|1.4|95.31|59.66|46.13, 7.12|
> |Llama-3.2-3B-Instruct|1.5|94.70|46.29|17.30, 5.08|
> |Qwen3-14B|1.8|97.49|6.98|23.52, 4.71|
> |Qwen3-32B|2.0|99.36|0.76|11.32|
>
> For the larger models we evaluated, it is possible to remove the TTP requirement. We verified that privacy protection can be achieved solely via the client's key, and analyzed the causes from both theoretical and experimental perspectives.
>
> >W3: Overclaiming "breaking the trilemma"
>
> We fully accept that "breaking" is too absolute and will revise it to "mitigating" or "navigating" the trilemma, bounded to a mitigation for large-model MaaS settings. We compare the best operating points of OSNIP against CAPE on the Privacy–Utility frontier (Utility = RP on arc_easy + piqa + mnli; Privacy = 1 − ASR; **BB** = black-box attacker, **WK** = white-box attacker with full encryptor weights but wrong key):
>
> |Model|Method|RP(%)|Privacy-BB(%)|Privacy-WK(%)|
> |-|-|-|-|-|
> |Llama-1B|CAPE (ε=3)|77.76|82.48|—|
> ||OSNIP|97.50|94.83|53.87|
> |Llama-3B|CAPE (ε=3)|70.77|82.14|—|
> ||OSNIP|99.31|98.08|82.70|
> |Qwen3-14B|CAPE (ε=10)|67.74|74.47|—|
> ||OSNIP|98.65|100.00|76.48|
> |Qwen3-32B|CAPE (ε=10)|69.91|75.27|—|
> ||OSNIP|99.74|100.00|88.68|
>
> The full Pareto curves are shown below:
>
> https://anonymous.4open.science/r/7z8k-7DF7/Privacy-Utility_Trade-off.png
>
> Even under the stronger WK threat model, OSNIP dominates CAPE on larger models, and the advantage grows with model scale—consistent with Corollary 2.6.
>
> >Q2: Formal privacy mechanism or empirical defense
>
> Our method is a formal privacy mechanism. We will state explicitly a quantitative indistinguishability result for the randomized OSNIP mechanism: for semantically equivalent inputs $h_1$ and $h_2$, the output distributions $P_1$ and $P_2$ satisfy $D_{\mathrm{TV}}(P_1,P_2)\le \frac{4}{\alpha}\exp(-\frac{(d-2)\epsilon^2}{2})$. This gives a $(0,\Delta)$-style guarantee with $\Delta=\frac{4}{\alpha}\exp(-\frac{(d-2)\epsilon^2}{2})$. We will add detailed evidence to the paper later.
>
> >Q3: Encryptor refresh under model updates
>
> For the vendor-trained deployment (W2), refresh is lightweight:
>
> |Model|Encryptor parameters(M)|Retrain Time|Hardware|
> |-|-|-|-|
> |Llama-3.2-1B|16.8|~17 min|4×A100|
> |Llama-3.2-3B-Instruct|37.78|~36 min|4×A100|
> |Qwen3-14B|104.91|~2 h|4×A100|
> |Qwen3-32B|104.91|~7 h|4×A100|
>
> For minor updates, direct transfer works well without retraining. Mean Retain is averaged over 5 tasks (arc_easy, hellaswag, piqa, mnli, sst2):
>
> |Updated Model|Update Type|Mean Retain(%)|
> |-|-|-|
> |DeepNews-LoRA-Qwen3-32B|LoRA SFT (0.8% params)|99.48|
> |Zhi-Create-Qwen3-32B|Full finetuning|96.81|
>
> Major upgrades are handled by cheap retraining (a one-time cost amortized across all clients); moderate updates often preserve utility without re-optimizing the encryptor.

---

> > ### Author Rebuttal · Reviewer_nqEZ · 2026-04-03
> >
> > Thanks so much for solving my concerns, thereby I raise the score to 4.

---

> > > ### Author Response · Authors · 2026-04-07
> > >
> > > We sincerely thank you for your engagement and for acknowledging that our responses have resolved your concerns. We are greatly encouraged by your feedback and deeply appreciate your re-evaluation and the updated score.
> > >
> > > If there are any additional questions or points you would like to discuss further, please feel free to share them. We remain available and would be more than happy to provide a detailed response.
> > >
> > > Thank you again for your valuable time and constructive efforts in helping us improve our work.

---

### Official Review · Reviewer_izbZ · 2026-03-10

**Soundness:** 3
**Presentation:** 3
**Significance:** 3
**Originality:** 3
**Overall Recommendation:** 4
**Confidence:** 4

**Summary:**

The authors propose OSNIP, a lightweight client-side encryption framework designed for privacy-preserving LLM inference in Model-as-a-Service (MaaS) settings. The core idea is to project original text embeddings into an "Obfuscated Semantic Null Space"—a high-dimensional region where the perturbed embeddings remain nearly orthogonal to the original ones (for privacy) but preserve the model’s output distribution (for utility). Furthermore, OSNIP employs a key-dependent stochastic mapping that synthesizes individualized perturbation trajectories unique to each user. Evaluations on 12 generative and classification benchmarks show that OSNIP achieves state-of-the-art performance, sharply reducing attack success rates while maintaining strong model utility under strict security constraints.

**Compliance With Llm Reviewing Policy:**

Affirmed.

**Key Questions For Authors:**

1. What is the performance of OSNIP on math and coding benchmarks, like HumanEval or MBPP.
2. How do you understand that in Table2, the llama model’s PPL increase more 30% than the Qwen model?
3. In the case of multi-turn conversations, does the "stochastic mapping" remain consistent, or does it change per turn?

**Limitations:**

1. Lack of the experiment on Math, coding benchmarks.

**Strengths And Weaknesses:**

Strengths
1. Define an “obfuscated semantic null space”, representing a high-dimensional region that preserves the model’s semantic distribution while remaining nearly orthogonal to the original embedding.
2. The defense is strong and the experiment is solid. Evaluations on 12 generative and classification benchmarks show that OSNIP achieves state-of-the-art performance, sharply reducing attack success rates while maintaining strong model utility under strict security constraints.
3. This paper introduces Obfuscated Semantic Null space Injection for Privacy (OSNIP), a lightweight client-side encryption framework. OSNIP uses geometric orthogonality as a primary privacy mechanism, employing a hinge-based constraint to minimize directional correlations.  This procedure projects perturbations by constraining them to remain approximately orthogonal to the underlying semantic invariant region. So, it can maintain the utility and protect the privacy.

Weaknesses
1. While OSNIP effectively secures embeddings through perturbation, a significant limitation lies in its potential to degrade performance on high-precision tasks such as mathematics and code generation. Because these domains are highly sensitive to even minor semantic shifts, the injection of stochastic noise may disrupt the logical integrity required for correct outputs. Furthermore, the absence of evaluations on math and coding benchmarks leaves the framework's robustness in these critical utility-driven areas unverified.
2. Didn’t report the training overhead of training an encryptor, especially the scenarios with lots of clients.
3. Hyperparameter Sensitivity: The "hinge-based constraint" likely requires careful tuning of the $\epsilon$ and $\delta$ parameters.

---

> ### Author Rebuttal · Authors · 2026-03-31
>
> We thank the reviewer for the insightful feedback.
>
> > W1, Q1: Performance on Math and Coding Benchmarks
>
> We conducted additional experiments on HumanEval and GSM8k:
>
> |Task|Model|Clean|OSNIP|Δ|
> |:-:|:-:|:-:|:-:|:-:|
> |HumanEval|Qwen3-14B|56.10%|60.37%|+4.27%|
> |GSM8k|Qwen3-14B|88.70%|88.48%|−0.22%|
> |HumanEval|Qwen3-32B|39.63%|35.37%|−4.26%|
> |GSM8k|Qwen3-32B|72.40%|67.78%|−4.62%|
>
> Qwen3-14B shows near-zero degradation on both benchmarks. For Qwen3-32B, we observe a modest drop of under 5 points on both tasks. These utility results are also consistent with our main privacy evaluation, where black-box KNN attack ASR for the corresponding Qwen3-14B/32B encryptors remains near zero. Notably, Qwen3-14B even improves on HumanEval (+4.27%), suggesting the perturbation can act as mild regularization.
>
> OSNIP preserves the logical integrity required for high-precision tasks such as math(GSM8k) and coding(HumanEval), while maintaining strong privacy.
>
> > W2: Training Overhead
>
> |Model|Encryptor parameters(M)|Training Time|Hardware|
> |:-:|:-:|:-:|:-:|
> |Llama-3.2-1B|16.8|~17 min|4×A100|
> |Llama-3.2-3B-Instruct|37.78|~36 min|4×A100|
> |Qwen3-14B|104.91|~2 h|4×A100|
> |Qwen3-32B|104.91|~7 h|4×A100|
>
> The encryptor is **shared at the model level, not the user level**: trained once per LLM deployment and distributed to all clients. Individual users incur zero training cost. At inference time, each user only generates a local random key and runs a forward pass through the shared encryptor, so overhead does not scale with the number of clients. This one-time cost is amortized over all future queries and users of the same deployment. Moreover, the encryptor remains lightweight relative to the base LLM, making client-side distribution practical.
>
> Training overhead is a one-time, deployment-level cost that does not scale with the number of clients.
>
> > W3: Hyperparameter Sensitivity
>
> We acknowledge that both the orthogonality margin ε (Eq. 15) and the separation margin δ (Eq. 19) influence the results, and smaller models exhibit greater sensitivity due to their limited null space capacity. To empirically assess robustness, we selected δ as a representative parameter and trained multiple encryptors under different values on Qwen3-14B and Qwen3-32B, where the richer obfuscated semantic null space (Corollary 2.6) allows the effect of parameter variation to be evaluated more cleanly:
>
> |Model|δ|RP(%)|KNN ASR BB (%)|
> |:-:|:-:|:-:|:-:|
> |Qwen3-14B|0.8|98.65|0.00|
> |Qwen3-14B|1.3|98.56|0.12|
> |Qwen3-14B|1.8|97.49|6.98|
> |Qwen3-32B|1.5|99.74|0.00|
> |Qwen3-32B|2.0|99.36|0.76|
>
> Here, **BB** denotes the **black-box** setting, i.e., the attacker only observes encrypted embeddings without access to the encryptor or key. OSNIP is stable within a practical range of δ: for Qwen3-14B, privacy degrades only when δ is pushed to 1.8; for Qwen3-32B, both RP and BB ASR remain robust across the tested values. We report δ as the representative margin; ε shows a similar qualitative trend. Full default hyperparameters will be provided in the appendix.
>
> OSNIP is practically robust to hyperparameter variation within a reasonable range, and larger models offer a more forgiving operating regime.
>
> > Q2: Why does Llama's PPL increase exceed 30% while Qwen's does not?
>
> This is consistent with Corollary 2.6 (Dimensionality Dividend). Qwen3-14B/32B provide a richer obfuscated semantic null space, allowing the encryptor to achieve strong privacy with smaller perturbation magnitude and lower PPL impact. Llama-3.2-1B/3B have a comparatively smaller null space, so achieving similar privacy requires larger perturbations and thus higher PPL. This trend supports our theoretical prediction that larger LLMs obtain a better privacy-utility trade-off under OSNIP.
>
> The PPL gap is directly explained by model scale — larger hidden dimensions yield a richer null space and lower perturbation cost, confirming the Dimensionality Dividend.
>
> > Q3: Multi-turn Stochastic Mapping
>
> In our multi-turn evaluation, we sample an **independent fresh random key per turn** and re-encrypt the full conversation history jointly at each turn. This reduces cross-turn linkability while preserving contextual coherence, since the entire history is encrypted together as one unit. We evaluated this on MT-Bench (80 two-round questions, Qwen3-32B):
>
> |Mode|Turn 1|Turn 2|Overall|
> |:-:|:-:|:-:|:-:|
> |Clean|6.83|7.09|6.96|
> |OSNIP|7.56|6.70|7.13|
>
> Per-turn re-encryption does not noticeably disrupt contextual coherence. Furthermore, our qualitative case study shows that when a user asks to update a JSON output from Turn 1 with new constraints in Turn 2, the model successfully integrates both turns' contexts. We will include this evaluation and the case study in the revised paper.
>
> Fresh-key per-turn re-encryption preserves multi-turn dialogue coherence with no observable quality loss.

---

> > ### Author Rebuttal · Reviewer_izbZ · 2026-04-03
> >
> > The review addresses most of my concerns. I believe this work clears the bar for acceptance, especially if the results presented during the rebuttal are incorporated into the paper.

---

> > > ### Author Response · Authors · 2026-04-07
> > >
> > > We sincerely thank you for your continued engagement and for acknowledging that our work clears the bar for acceptance. We are greatly encouraged by your positive feedback. We will ensure that all new results, analyses, and clarifications presented during this rebuttal phase are thoroughly incorporated into the final version of the paper.
> > >
> > > We understand from your assessment that you may have some follow-up questions. If you have any remaining questions, specific concerns, or points you would like to discuss further, please feel free to share them. We remain available and would be more than happy to provide a detailed response.
> > >
> > > Thank you again for your time and constructive efforts in helping us improve our work.

---

### Official Review · Reviewer_CZQ4 · 2026-03-12

**Soundness:** 3
**Presentation:** 3
**Significance:** 3
**Originality:** 3
**Overall Recommendation:** 3
**Confidence:** 3

**Summary:**

The manuscript discusses privacy-preserving inference for Large Language Models (LLMs) under the Model-as-a-Service (MaaS) paradigm, where existing solutions are trapped in a rigid privacy-utility-efficiency trilemma and treat the high dimensionality of LLM embeddings as a burden rather than an advantage. This paper intends to explore the key question of how to leverage the over-parameterization of LLMs to achieve lightweight, denoising-free privacy protection without sacrificing model utility or requiring server-side modifications.

To address this, the authors first formally define the Obfuscated Semantic Null Space, a high-dimensional region that preserves the model's predictive distribution while enforcing near-orthogonality to the original input embedding. They theoretically prove the non-emptiness of this space under mild conditions. Based on this theory, the authors propose OSNIP, a lightweight client-side encryption framework that injects perturbations along model-insensitive directions to project clean embeddings into the obfuscated semantic null space. OSNIP further incorporates a key-conditioned personalized perturbation mechanism to reduce cross-user linkability, and a utility-gated curriculum learning strategy to stabilize multi-objective optimization.

**Compliance With Llm Reviewing Policy:**

Affirmed.

**Key Questions For Authors:**

1.In the main text, the authors explain the function of key hyperparameters including the orthogonality margin, loss weights, separation margin, and safe interval, but do not provide their default values, tuning ranges, or selection principles. Meanwhile, there is a notation inconsistency in Equation (20): the text mentions β controls the strength of key personalization, yet β does not appear in the formula itself. Could the authors please clarify the correct notation for the weight of the diversity loss, and provide the default values, tuning guidelines, and selection principles of all the aforementioned core hyperparameters for different model scales?
2.The authors introduce a user-specific secret key k for personalized perturbation, but do not provide formal theoretical analysis or empirical validation on the security boundary of this key mechanism. Could the authors please elaborate on the security requirements for the key, and provide theoretical or experimental results on how privacy protection performance degrades when the key is leaked or compromised?
3.The authors note that the work is motivated by prior studies leveraging the null space for LLM knowledge editing and safety enhancement. However, the paper does not clearly articulate the essential theoretical extensions to the null space framework. Could the authors please explicitly clarify the key differences in the theoretical definition, constraint conditions, and mathematical properties of the obfuscated semantic null space, relative to the null space used in prior model editing and safety alignment works?

**Limitations:**

yes

**Strengths And Weaknesses:**

1. Soundness
**Strengths**
- Rigorous theoretical foundation. The authors formally define core concepts including the semantic null space, geometric obfuscation region, and obfuscated semantic null space. They provide complete mathematical proofs for the existence theorem of the null space and the asymptotic semantic dominance corollary in the appendix, with reasonable and mild assumptions.
- Comprehensive and rigorous experimental design. The evaluation covers 4 mainstream LLMs with scales from 1B to 32B, and compares against 3 latest SOTA baselines (Cape, DYNTEXT, InferDPT) with their official optimal configurations. The benchmarks span closed-ended reasoning/classification tasks, open-ended text completion, and summarization. Privacy is evaluated on two mainstream embedding inversion attacks (KNN and vocabulary-matching), with additional ablation studies on white-box adaptive attacks, convergence analysis, internal mechanism visualization, latency testing, and case studies.

**Weaknesses**
- Insufficient security analysis of the key mechanism. The paper introduces a user-specific secret key k for personalized perturbation, but does not formally analyze the security boundary of the key or the degradation of privacy protection when the key is leaked, with theoretical or experimental support.

2. Presentation
**Strengths**
- The paper follows a logical flow from problem setup, theoretical analysis, framework design, experimental validation and supplementary materials.

**Weaknesses**
- Notation inconsistency. Equation (20) mentions that β controls the strength of key personalization, but β does not appear in the formula itself, which is a typo needing correction.
- Insufficient explanation of hyperparameter selection. The text explains the function of key hyperparameters (e.g., orthogonality margin, loss weights, separation margin, safe interval) but does not provide their default values or selection principles, which are critical for reproducibility.

3. Significance
**Strengths**
- The work addresses a key challenge in MaaS privacy protection. Existing privacy-preserving LLM inference methods face a persistent trade-off between privacy, utility, and inference efficiency. Cryptographic schemes such as HE and MPC often introduce considerable computational overhead, differential privacy-based perturbation methods tend to compromise semantic integrity and model utility, and most existing solutions require non-trivial server-side model modifications or additional post-processing/denoising steps. OSNIP effectively mitigates this trade-off, realizing lightweight client-side encryption with no server-side modifications, near-lossless model utility, robust defense against embedding inversion attacks, and negligible inference overhead, offering a practically valuable solution for real-world MaaS deployment.
- The work extends the null space framework to the field of LLM privacy protection. While prior works have leveraged the null space concept for LLM knowledge editing and safety enhancement, this paper is the first to adapt and formalize this framework for the privacy-preserving inference task. Building on insights from these related works, the authors demonstrate that the over-parameterization of LLMs gives rise to an Obfuscated Semantic Null Space—a high-dimensional region that preserves the model's output distribution while achieving geometric decorrelation from the original embedding, and formalize its mathematical properties for privacy protection. This work verifies the broader applicability of the null space framework across different LLM research directions, and establishes a theoretically grounded, denoising-free privacy protection paradigm, providing a new technical reference for subsequent lightweight LLM privacy research.

**Weaknesses**
- Limited applicability in mainstream closed-source MaaS scenarios. OSNIP requires a trusted third party to have gradient access to the server-side LLM for training the encryptor, which is not feasible for closed-source MaaS APIs (e.g., OpenAI, Anthropic) that do not expose model gradients. The paper does not discuss adaptation schemes for this pure black-box setting, limiting its real-world deployment scope.

4. Originality
**Strengths**
- Novel core concept. This work formally defines the obfuscated semantic null space tailored to LLM privacy-preserving inference, and provides theoretical proofs for its existence and the associated dimensionality dividend. This offers a distinct counterperspective to the conventional view in the privacy domain that treats the high dimensionality of LLM embeddings as a burden, rather than an asset for protection.
- Valuable empirical and theoretical insight. The paper validates the privacy protection dividend of high-dimensional LLM representations through both theory and experiments, demonstrating that larger models can achieve an improved privacy-utility trade-off — a trend that contrasts with the existing baseline methods. This finding provides a fresh perspective for subsequent research on large model privacy protection.

**Weaknesses**
- Insufficient distinction from prior null space work. The null space concept has been explored in prior model editing and safety alignment work. While the application to privacy protection is novel, the paper could more clearly articulate the essential theoretical extensions it makes to the null space framework, compared to these prior works.

---

> ### Author Rebuttal · Authors · 2026-03-31
>
> We thank the reviewer for the insightful feedback.
>
> > W1, Q2: Key mechanism
>
> Thanks for highlighting that. The key in OSNIP is the sole source of randomness in the client-side mapping. Each time a query is made, users can generate a new key, and the encryptor then maps the clean embedding to one point inside $\mathcal N$ conditioned on that key. Therefore, the relevant security question is not if an adversary can better characterize that feasible obfuscated region after observing many ciphertexts, but if such observations allow the adversary to invert them back to the original clean embedding uniquely.
>
> The result is negative given fresh independent hidden keys. Multiple ciphertexts per input merely yield redundant samples within the same feasible region. After intercepting m ciphertexts, the adversary’s remaining uncertainty set still satisfies $\sigma(A_m(X)) \ge \alpha - 2m \exp(-\frac{(d-2)\epsilon^2}{2}),$ where $\alpha$ denotes the semantic coverage rate. Hence, the privacy leakage per additional query is upper bounded by an exponentially small term $2\exp(-(d-2)\epsilon^2/2)$, and the number of queries required to exhaust this uncertainty grows exponentially with the representation dimension. That is, driving this lower bound to zero requires a critical number of intercepted queries $m_{crit} \ge \frac{\alpha}{2}\exp(\frac{(d-2)\epsilon^2}{2})$, which scales exponentially with the ambient dimension. Intuitively, observing more keyed samples helps reveal the geometry of the obfuscated region, but does not by itself collapse the inverse problem to a unique clean embedding. In the revised version, we will add the complete proof.
>
> > W2, Q1: Notation
>
> We appreciate that. In the revised manuscript, we will clarify that $\lambda_1$ and $\lambda_2$ govern privacy preservation and key personalization intensity, respectively. These changes will be incorporated into the updated version.
>
> > W3, Q1: Hyperparameter
>
> The following table shows the specific values of some hyperparameters in the article:
>
> |Model|Orth & Sep margin(cos <x, L2>y)|Loss Weight|Safe interval|
> |-|-|-|-|
> |Llama-1B|0.4,0.4|Util 1.0, Priv 1.0, Div 2.0|0.05-0.15|
> |Llama-3B|0.4,0.5|⬆️|⬆️|
> |Qwen3-14B|0.35,0.8|⬆️|⬆️|
> |Qwen3-32B|0.25,1.5|⬆️|⬆️|
>
> Most hyperparameters remain stable across model scales and do not require retuning when the backbone changes. In practice, only a small subset of geometry-related margins, especially the orthogonality margin $\epsilon$ (Eq. 15) and the separation margin $\delta$ (Eq. 19), need mild empirical adjustment.
>
> Our rule of thumb is that $\delta$ generally increases with model dimensionality, while $\epsilon$ generally decreases with model dimensionality, reflecting that larger LLMs admit a richer obfuscated semantic null space.
>
> We will add the full selection principles and detailed hyperparams to the updated manuscript text.
>
> > W4: Close-sourced LLMs
>
> This paper proposes a privacy-preserving solution in which the encryptor is trained by a TTP when the model is small or the client is computationally constrained. When the model is sufficiently large, experiments indicate that the MaaS provider can train the encryptor once and distribute it to clients, while protection at inference time remains on the client side through secret keys.
>
> Tab 7 and the below show that on Qwen3-32B, OSNIP remains secure under the wrong-key setting against white-box attackers. We analyze the causes of this phenomenon from a theoretical (Theorem 2.5) and experimental(Fig 4) view.
>
> | Model|Method|Setting|RP(%)| ASR BB(%)|ASR WB(%)|
> |:-|:-|:-|:-|:-|:-|
> |Qwen-32B|Our|key_l2=1.5|99.74|0.00|23.71|
> |||key_l2=2.0|99.36|0.76|11.32|
>
>
> > W5, Q3: Distinction
>
> Thanks. Prior works are inspirations, but their null spaces are algebraic and layer-specific; OSNIP instead defines the null space at the level of the frozen nonlinear predictor $f_\theta$ around a reference embedding $h$.
>
> Constraint conditions: prior work constrains an internal perturbation operator to vanish on preserved knowledge or benign activations, whereas OSNIP constrains the perturbed embedding itself to satisfy both utility tolerance $\delta$ and privacy-specific orthogonality margin $\epsilon$. This is the key extension for privacy-preserving inference, where the protected object is the released input representation rather than model parameters or a refusal steering map.
>
> Mathematical properties: prior work mainly establishes exact preservation or null-space equivalence in linear internal modules. OSNIP instead proves a high-dimensional existence result: Theorem 2.5 gives a sufficient condition for non-emptiness, and Cor 2.6 shows that the orthogonality constraint becomes asymptotically non-binding as $d$ grows. In the idealized linear case in Sec. 2, our function-level definition reduces to the classical algebraic null-space/kernel view, so OSNIP contains that setting as a special case. We will add this comparison in the theory/related-work discussion.

---

> > ### Author Rebuttal · Reviewer_CZQ4 · 2026-04-03
> >
> > The rebuttal addresses several of my concerns, but not all of them fully. In particular, the authors provided concrete hyperparameter values and practical selection guidelines, which improves reproducibility, and they also clarified the intended deployment path in a provider-cooperative closed-source setting. The rebuttal further gives a clearer explanation of the threat model and the role of fresh randomized keys. However, I still have some remaining concern regarding originality. While the authors now explain the differences between OSNIP and prior null-space-based work more clearly, yet I am not fully convinced that the paper sufficiently establishes the extent of its essential theoretical novelty relative to existing null-space formulations used in model editing and safety alignment. In my view, the application to privacy-preserving inference is interesting and potentially valuable, but the paper would benefit from a sharper articulation of what is fundamentally new in the theory, beyond adapting the null-space perspective to this setting. Therefore, I consider my concerns to be only partially resolved.

---

> > > ### Author Response · Authors · 2026-04-07
> > >
> > > We thank the reviewer for acknowledging our prior clarifications. To address the remaining concern, we compare OSNIP with null-space formulations across five subfields.
> > >
> > > **Null-Space Literature Survey.**
> > >
> > > Prior works share a common mathematical paradigm:
> > >
> > > *Knowledge Editing & Safety Alignment.* AlphaEdit [1] projects weight perturbations $\Delta$ into the null space of preserved-knowledge keys $K\_0$ via SVD-based projection, enforcing $\Delta' K\_0 = 0$. AlphaSteer [2] constrains a learnable steering matrix to the left null space of benign activations $H\_b$, ensuring $\Delta H\_b = 0$.
> > >
> > > *Fairness.* INLP [3] iteratively projects representations onto the null space $N(W) = \\{x \mid Wx = 0\\}$ of linear classifiers trained to predict protected attributes, removing linearly extractable bias.
> > >
> > > *Continual Learning.* Adam-NSCL [4] projects parameter updates into the approximate null space of old-task input-feature covariance matrices via SVD. VPT-NSP [5] constrains prompt updates $\Delta P$ to satisfy $Q\_{X\_t} W\_k^\top \Delta P^\top = 0$, preserving ViT attention for previous tasks.
> > >
> > > *Machine Unlearning.* UNSC [6] projects unlearning gradients into the null space of remaining-data features $R\_l$ at each layer, ensuring $\langle \Delta w^l, r\_i^l \rangle = 0$.
> > >
> > > *Parameter-Efficient Tuning.* LoRA-Null [7] initializes LoRA matrices so that $BAX\_{\text{pre}} \approx 0$, placing low-rank updates in the null space of pre-trained input activations.
> > >
> > > **Unified paradigm.** Despite spanning five areas, all works share a unified structure: finding the algebraic kernel of a deterministic linear operator ($Mx = 0$) within internal modules. The null space is always a linear subspace, and existence follows from the Rank-Nullity Theorem.
> > >
> > > **How OSNIP Fundamentally Differs.**
> > >
> > > *Dimension 1: Definition Level.*
> > > All prior works operate within one or a few linear internal layers/modules, solving $Mx = 0$ for weights, activations, or gradients. OSNIP defines its null space at the level of the entire frozen non-linear predictor $f\_\theta$ (i.e., LLMs), on the released input representation. The constraints become metric-geometric inequalities:
> > > $d\_{\mathcal{P}}(f\_\theta(\mathbf{h}), f\_\theta(\mathbf{z})) \le \delta$ and $|\cos(\mathbf{h}, \mathbf{z})| \le \epsilon$.
> > > The solution set is not a linear subspace but a topological neighborhood on a non-linear manifold — a qualitatively different mathematical object. In the idealized linear case in Sec. 2, our function-level definition reduces to the classical algebraic null-space/kernel view, so OSNIP contains that setting as a special case.
> > >
> > > *Dimension 2: Existence Proof.*
> > > Prior null-space existence is a direct corollary of the Rank-Nullity Theorem (more parameters than constraints implies non-trivial kernel) — standard linear algebra. OSNIP Theorem 2.5 confronts a harder problem: orthogonal perturbations can easily destroy non-linear semantic mappings, so linear-algebraic reasoning is insufficient. We establish non-emptiness via exponential concentration inequalities from high-dimensional probability, providing a statistical bound on the semantic coverage rate $\alpha$. This theoretical conclusion is qualitatively absent from all surveyed works.
> > >
> > > *Dimension 3: Dimensionality Dividend.*
> > > Prior works only determine "how many dimensions remain available for projection" without studying the asymptotic relationship between dimensionality and non-linear objectives. OSNIP Corollary 2.6 is the first to prove that as $d \to \infty$, the orthogonal privacy constraint becomes asymptotically non-binding in the non-linear semantic distribution — the privacy-utility conflict theoretically vanishes for larger LLMs. This theorem type does not exist in any prior null-space work.
> > >
> > > **Summary.** OSNIP transitions null-space methodology from algebraic constraint solving on linear internal modules to geometric metric topology on non-linear latent distributions. Theorem 2.5 (non-linear non-emptiness) and Corollary 2.6 (dimensionality dividend) are independent theoretical contributions whose conclusions are absent from all seven surveyed works. This is a mathematical framework generalization, not application adaptation. We hope this comparison fully establishes the theoretical novelty of our work.
> > >
> > > [1] Fang et al., AlphaEdit: Null-Space Constrained Knowledge Editing for Language Models, ICLR 2025.
> > >
> > > [2] Sheng et al., AlphaSteer: Learning Refusal Steering with Principled Null-Space Constraint, ICLR 2026.
> > >
> > > [3] Ravfogel et al., Null It Out: Guarding Protected Attributes by Iterative Nullspace Projection, ACL 2020.
> > >
> > > [4] Wang et al., Training Networks in Null Space of Feature Covariance for Continual Learning, CVPR 2021.
> > >
> > > [5] Lu et al., Visual Prompt Tuning in Null Space for Continual Learning, NeurIPS 2024.
> > >
> > > [6] Chen et al., Machine Unlearning via Null Space Calibration, IJCAI 2024.
> > >
> > > [7] Tang et al., Put the Space of LoRA Initialization to the Extreme to Preserve Pre-trained Knowledge, AAAI 2026.

---

### Official Review · Reviewer_DFZi · 2026-03-12

**Soundness:** 3
**Presentation:** 3
**Significance:** 3
**Originality:** 3
**Overall Recommendation:** 4
**Confidence:** 4

**Summary:**

This paper introduces OSNIP, a privacy-preserving framework for user input in cloud-based LLM inference scenarios. The core concept leverages the "semantic null space" inherent in the high-dimensional embedding space of large models—regions abundant with vectors that are geometrically orthogonal yet semantically equivalent to the original embeddings.  The study trains a lightweight encryptor that, in conjunction with a user-specific private key, maps the original prompt embeddings into the semantic null space. This ensures that the server cannot reconstruct the original input, while the LLM remains capable of producing accurate outputs. The private key mechanism guarantees white-box security, rendering the input indecipherable even if an attacker gains full access to the encryptor's parameters.  Experimental results demonstrate that this approach reduces attack success rates to nearly zero, while preserving almost perfect model utility. Furthermore, the additional computational overhead on the client side is minimal, underscoring the method's practicality and robustness.

**Compliance With Llm Reviewing Policy:**

Affirmed.

**Key Questions For Authors:**

* The paper experimentally observes the existence of the semantic null space but does not provide a theoretical proof. Does the dimension and size of this space vary significantly with changes in model architecture or training data? If certain models have a very small semantic null space, would the method fail?
* The paper assumes a "generic attacker." If the attacker is aware of the use of OSNIP and designs specific attack strategies targeting the geometric structure of the semantic null space, would the current security guarantees still hold?
* If an attacker accumulates $N$ encrypted embeddings from the same user, how does the success rate of correlation attacks grow with $N$? Is there a safe usage limit for the number of queries?
* Privacy protection for single queries is relatively straightforward, but real-world usage involves continuous conversations where semantic dependencies exist between turns. Would independent encryption for each turn disrupt contextual coherence? How can a balance be achieved between privacy protection and dialogue quality?
* The paper assumes that the client can run the encryptor, but the encryptor itself is a neural network. On mobile devices or IoT platforms, is the computational overhead truly acceptable?
* Differential privacy protects privacy by injecting noise, while OSNIP relies on space mapping. The formalization of privacy guarantees differs significantly between the two. Can OSNIP provide a quantifiable privacy guarantee similar to $\epsilon$-differential privacy, rather than relying solely on empirical attack success rate metrics?

**Limitations:**

yes

**Strengths And Weaknesses:**

**Strengths:**

- Transforms the mathematical property of "redundant directions in the high-dimensional model space" into a practical privacy-preserving tool, offering a unique perspective.
- Encryption is performed entirely on the client side, requiring no modifications to the server-side LLM, which minimizes deployment challenges.
- In the tested scenarios, attackers gain virtually no useful information, while the quality of the model's responses remains almost unaffected, achieving a balance with no significant trade-offs.
- Even if attackers obtain the complete parameters of the encryptor, they cannot decrypt the input without the private key, making this approach more robust than many similar solutions.

**Weaknesses:**

- OSNIP does not protect the privacy of the output results, making it unsuitable for scenarios where the output contains sensitive information.
- The security analysis is based on single-query scenarios, and the paper does not address the risk of attackers conducting correlation analysis by accumulating multiple records when users use the same private key over time.
- The experiments only test general attack methods and do not design adaptive attacks specifically targeting the OSNIP mechanism, which may lead to overly optimistic security conclusions.

---

> ### Author Rebuttal · Authors · 2026-03-31
>
> We thank the reviewer for the insightful feedback.
>
> > W1: Output privacy boundary
>
> We agree that OSNIP specifically targets the privacy of input embeddings rather than generated outputs; this threat model is also consistent with the dominant setting in prior privacy-preserving LLM inference literature, which primarily protects the user's prompt, while treating output-side leakage as an orthogonal challenge that can be handled by downstream filtering.
>
> >  W2, Q3: Multi-query/repeated-query attacks
>
> We clarify that OSNIP is a randomized map conditioned on keys: users may switch or update it at any time. Hence, repeated encryptions of the same plaintext are independent randomized instances, not repeated observations under a static projection.
>
> To address this concern, we additionally evaluated repeated-query aggregation attacks, in which an adversary averages N ciphertexts of the same plaintext. We considered two representative geometric aggregations:
>
>
> |N|naive_avg_Top1|Top5|sphere_avg_Top1|Top5|
> |-|-|-|-|-|
> |1|13.02%|16.21%|13.04%|16.21%|
> |1000|10.67%|11.64%|13.93%|16.72%|
> |100000|10.65%|11.57%|13.93%|16.54%|
>
>
> As these results show, we do not observe monotonic privacy degradation when N increases, which suggests that, under fresh-key randomization, accumulating more ciphertexts does not steadily strengthen these attacks.
>
> > W3, Q2: Adaptive attacks targeting OSNIP
>
> When attackers know OSNIP and have white-box access to the encryptor, privacy can still be effectively protected via a client-side random key. Appendix D.1 / Table 7 supports this point.
> We also evaluated KNN-based adaptive white-box (WB) attacks across 4 models, and CAPE's black-box (BB) results as reference:
>
>
> |Model|Method|Setting|RP(%)|ASR BB(%)|ASR WB(%)|
> |-|-|-|-|-|-|
> |Llama-1B|CAPE|eps=3~14|77.25~83.52|17.52~32.29|-|
> ||Our|key_l2=0.4|95.31|5.17|100|
> |||key_l2=1.9|95.20|58.81|52.86|
> |Llama-3B|CAPE|eps=3~14|70.77~79.45|17.86~33.69|-|
> ||Our|key_l2=0.5|99.31|1.92|46.55|
> |||key_l2=1.0|94.70|38.22|42.34|
> |Qwen-14B|CAPE|eps=3~14|62.00~74.42|16.63~32.89|-|
> ||Our|key_l2=0.8|98.65|0.00|91.76|
> |||key_l2=1.3|98.56|0.12|45.45|
> |Qwen-32B|CAPE|eps=3~14|64.23~77.71|16.88~32.31|-|
> ||Our|key_l2=1.5|99.74|0.00|23.71|
> |||key_l2=2.0|99.36|0.76|11.32|
>
>
> Consistent with Theorem 2.5, our results suggest a trade-off between black and white-box attacks when the hidden dimension is small, so we will revise the wording to avoid overstating security. This tension is largely alleviated as the dimension increases. Notably, compared with CAPE's standard BB setting, OSNIP still achieves substantially higher RP. We will revise the manuscript to clarify this.
>
> > Q1: Dependence of $N\_{\delta,\epsilon}$
>
> We respectfully clarify that our claim is not merely empirical: Theorem 2.5 already provides a sufficient condition for the non-emptiness of $N\_{\delta,\epsilon}(h)$. Since $N\_{\delta,\epsilon}(h)\subset R^d$, what varies across models is not the ambient dimension but the effective feasible size, which depends on $d$ and the semantic coverage rate $\alpha\_\delta(\mathbf h)$. Different architectures may affect $\alpha\_\delta(h)$, but experiments show that our encryptor exhibits generalization even after model updates (See Response to nqEZ Q3). Our theoretical point is that once $\alpha\_\delta(h)$ is non-negligible, larger $d$ makes orthogonal obfuscation exponentially more favorable. This matches our results: smaller models have a weaker privacy-utility trade-off, but OSNIP remains effective even at 1B. Thus, a smaller $N\_{\delta,\epsilon}$ makes optimization harder, rather than causing outright failure.
>
> > Q4: Multi-turn dialogue coherence
>
> Under a fresh random key, while leaving the server-side autoregressive inference pipeline unchanged, we evaluate this setting on MT-Bench, using the DeepSeek API as the judge.
>
>
> |MT-Bench|Turn 1|Turn 2|Overall|
> |-|-|-|-|
> |Clean|6.83|7.09|6.96|
> |Our|7.56|6.70|7.13|
>
>
> We will include the detailed benchmark breakdown and representative case studies in the revised paper.
>
> > Q5: Client overhead / mobile deployment
>
> Appendix D.5, Table 4 in the paper supports low additional latency relative to prior baselines. The client-side encryptor is an adapter-style module, with the following parameter sizes:
>
>
> |Base model|Encryptor parameters(M)|
> |-|-|
> |Llama-3.2-1B|16.8|
> |Llama-3.2-3B-Instruct|37.78|
> |Qwen3-14(32)B|104.91|
>
>
> These results support that OSNIP introduces very low additional latency compared with prior baselines, while remaining post-processing free.
>
> > Q6: Quantifiable privacy guarantee
>
> We will state explicitly a quantitative indistinguishability result for the randomized OSNIP mechanism: for semantically equivalent inputs $h\_1$ and $h\_2$, the output distributions $P\_1$ and $P\_2$ satisfy $D_{\mathrm{TV}}(P\_1,P\_2)\le \frac{4}{\alpha}\exp(-\frac{(d-2)\epsilon^2}{2})$. This gives a $(0,\Delta)$-style guarantee with $\Delta=\frac{4}{\alpha}\exp(-\frac{(d-2)\epsilon^2}{2})$. We will add detailed evidence to the paper later.

---

> > ### Author Rebuttal · Reviewer_DFZi · 2026-04-03
> >
> > Thank you to the authors for their detailed responses and for providing supplementary experimental results. My concerns have been resolved, and I will maintain my positive stance.

---

> > > ### Author Response · Authors · 2026-04-07
> > >
> > > We sincerely thank you for your time and constructive feedback. We are greatly encouraged to hear that your concerns have been fully resolved and that you are maintaining your positive stance on our work.
> > >
> > > While your concerns have been adequately addressed, if you have any additional questions or points you would like to discuss further, please feel free to share them. We remain available and would be more than happy to provide a detailed response.
> > >
> > > Thank you again for your continued support and engagement.

---

### Decision · Program_Chairs · 2026-04-30

**Decision:**

Accept (regular)

**Comment:**

Overall, the reviewers find this paper to be technically solid, with a novel perspective on privacy-preserving LLM inference by leveraging high-dimensional latent space structure. They consistently acknowledge the strong experimental performance across diverse setups as well as a rigorous theoretical foundation. At the same time, Reviewer CZQ4 raises concerns about the theoretical novelty relative to existing null-space formulations used in model editing and safety alignment. In addition, the assumptions for theorems may not hold across diverse practical scenarios. The authors' rebuttal addressed many of these issues with additional ablation analyses and clarifications to revise overstated claims. On balance, I recommend a weak accept, as the paper provides a technically solid idea.